



# A Joint Reconstruction and Model Selection Approach for Large Scale Inverse Modeling

Malena Sabaté Landman[1], Julianne Chung[1], Jiahua Jiang[2], Scot M. Miller[3], and Arvind K. Saibaba[4]

[1]Department of Mathematics, Emory University, Atlanta, GA, USA
[2]School of Mathematics, University of Birmingham, Birmingham, UK
[3]Department of Environmental Health and Engineering, Johns Hopkins University, Baltimore, MD, USA
[4]Department of Mathematics, North Carolina State University, Raleigh, NC, USA

**Correspondence:** Julianne Chung (jmchung@emory.edu)

**Abstract.** Inverse models arise in various environmental applications, ranging from atmospheric modeling to geosciences. Inverse models can often incorporate predictor variables, similar to regression, to help estimate natural processes or parameters of interest from observed data. Although a large set of possible predictor variables may be included in these inverse or regression models, a core challenge is to identify a small number of predictor variables that are most informative of the model, given limited observations. This problem is typically referred to as model selection. A variety of criterion-based approaches are commonly used for model selection, but most follow a two-step process: first, select predictors using some statistical criteria, and second, solve the inverse or regression problem with these predictor variables. The first step typically requires comparing all possible combinations of candidate predictors, which quickly becomes computationally prohibitive, especially for large-scale problems. In this work, we develop a one-step approach, where model selection and the inverse model are performed in tandem. We reformulate the problem so that the selection of a small number of relevant predictor variables is achieved via a sparsity-promoting prior. Then, we describe hybrid iterative projection methods based on flexible Krylov subspace methods for efficient optimization. These approaches are well-suited for large-scale problems with many candidate predictor variables. We evaluate our results against traditional, criteria-based approaches. We also demonstrate the applicability and potential benefits of our approach using examples from atmospheric inverse modeling based on NASA's Orbiting Carbon Observatory 2 (OCO-2) satellite.

## 1 Introduction

Inverse modeling is used across the Earth sciences, engineering, and medicine to estimate a quantity of interest when there are no direct observations of that quantity, but rather, there are only observations of related quantities (e.g., Tarantola, 2005; Nakamura and Potthast, 2015). For example, inverse modeling is used in contaminant source identification to estimate the sources of pollution when the only observations available are downwind or downstream measurements of pollution concentrations. In hydrology, inverse modeling can be used to estimate hydraulic conductivity and storativity using pumping tests. Similarly, seismic tomography is an inverse modeling technique for understanding the subsurface of the Earth using seismic waves. In





this paper, we consider applications from atmospheric inverse modeling but the approaches we discuss are applicable across

the environmental sciences.

In many applications of inverse modeling, multiple data sources can be used as prior knowledge to help predict the unknown quantity. For example, in environmental inverse problems, there are increasing numbers of satellite sensors that provide detailed data on both the built and natural environment and can serve as predictors of pollution emissions, or there may be multiple models that provide different predictions of the unknown quantity. However, the increasing availability of prior information

or predictor variables for inverse modeling can be both a blessing and a curse. On one hand, the existence of more predictor variables can help increase the accuracy of the posterior estimate. On the other hand, the existence of so many predictor variables or prior information can necessitate difficult choices about which to include or exclude from the inverse model.

Throughout this manuscript, we draw upon case studies from atmospheric inverse modeling (AIM) to highlight the challenges of handling multiple predictor variables. In AIM, the primary goal is to estimate surface fluxes of a greenhouse gas

or air pollutant using observations of gas mixing ratios in the atmosphere collected from airplanes, TV towers, or from satellites. A model of atmospheric winds is usually required to quantitatively link surface fluxes with downwind observations in the atmosphere (e.g., Brasseur and Jacob, 2017; Enting, 2002). AIM epitomizes the challenges posed by numerous predictor variables since there is a wide range of available data from multiple sources that can be used as predictors. A common choice is to use a biogeochemical, process-based, or inventory model of the fluxes. For some air pollutants or greenhouse gases, there

is a plethora of available models to choose from. For example, the most recent Global Carbon Project report on the global $CO_2$ cycle includes $CO_2$ flux predictions from 16 biogeochemical flux models, and the most recent methane ($CH_4$) report includes 13 biogeochemical models of $CH_4$ fluxes from global wetlands (Saunois et al., 2020; Friedlingstein et al., 2022). In theory, any of these models could be used within the prior to help predict either $CO_2$ or $CH_4$ fluxes using AIM. In some studies, rather than using a biogeochemical model prediction of fluxes, the authors use environmental variables to predict natural $CO_2$ and $CH_4$

fluxes, including estimates of soil moisture, air temperature, etc. (e.g., Gourdji et al., 2008, 2012; Miller et al., 2014, 2016; Randazzo et al., 2021; Chen et al., 2021a, b). This approach is often referred to in the AIM literature as geostatistical inverse modeling (Michalak et al., 2004). Some modelers also use remote sensing products like indicators of vegetation greenness and estimates of soil inundation (e.g., Shiga et al., 2018a, b; Zhang et al., 2023). Modelers have further used this approach as a means to evaluate the relationships between these environmental variables and $CO_2$ or $CH_4$ fluxes (e.g., Fang and Michalak,

2015; Chen et al., 2021b; Randazzo et al., 2021).

One solution to this challenge is to choose a single predictor variable or a single biogeochemical model to construct the prior of the inverse model, as is common in the existing inverse modeling literature (e.g., Brasseur and Jacob, 2017; Tarantola, 2005). The choice of predictor variable might be based on the modeler's expert knowledge or personal preference. Geostatistical studies, by contrast, often assimilate multiple predictor variables simultaneously to predict the unknown quantity of interest.

Suppose that one has identified a set of $p$ predictor variables or covariates that may help predict the quantity of interest ($s$), and these covariates have been assembled into a matrix $\mathbf{X} \in \mathbb{R}^{n \times p}$ where each column contains a different predictor variable (e.g. Kitanidis and VoMvoris, 1983; Michalak et al., 2004). That is, we consider the model,

$$s = \mathbf{X}\boldsymbol{\beta} + \boldsymbol{\zeta}. \tag{1}$$





where $\mathbf{s} \in \mathbb{R}^n$ is a vector representing the unknown quantity of interest (e.g., spatial or spatiotemporal maps of emissions of pollution), $n$ indicates the total number of unknowns to estimate (e.g, at different locations and at different times), and $\boldsymbol{\zeta} \in \mathbb{R}^n$ is referred to here as the stochastic component. It captures variability in the unknown quantity that is not described by the predictor variables. In this setup, the stochastic component is estimated as part of the inverse model along with the set of coefficients or scaling factors $\boldsymbol{\beta} \in \mathbb{R}^p$ that describe the relationships between the predictor variables and the quantity of interest ($\mathbf{s}$). We use the formulation in Eq. (1) throughout this manuscript because it affords more flexibility to incorporate a greater number of predictor variables.

This framework facilitates the use of more than one predictor variable, but the inclusions of all possible predictors within an inverse model has numerous pitfalls. First, many of the biogeochemical models or environmental variables described above are colinear, meaning that they are highly correlated with one another. Colinearity can cause numerous problems in linear modeling because the model has difficulty determining how to weight very similar or non-unique predictor variables (e.g., Kutner et al., 2004). There are several available metrics to identify colinear variables; one can examine the correlations between pairs of predictors or calculate variance inflation factors, among other metrics (e.g., Kutner et al., 2004). Second, the inclusion of all available predictors can cause the inverse model to over-fit available observations. The inclusion of more predictor variables in a linear model will always improve the model-data fit. In fact, one can perfectly predict $n$ observations by using $n$ predictor variables, yielding a model-data fit of $R^2 = 1$, but there are dangers of including too many predictors or covariates in linear modeling. The problem of over-fitting is reviewed in Zucchini (2000).

Model-selection techniques have been considered for obtaining important and relevant predictors in inverse problems. These techniques include the partial F-test and Bayesian Information Criterion (BIC), among other model selection methods (see Sect. 2; e.g., Gourdji et al., 2008, 2012; Yadav et al., 2016). However, these approaches can be computationally expensive, especially for large-scale problems with many candidate predictors.

### 1.0.1 Summary of challenges and contributions.

Making an informed decision about which predictor variables or prior information to incorporate or discard in the inverse model is important yet very challenging, as it is often not straightforward to distinguish between informative and non-informative variables. In addition, choosing a subset of $s$ predictor variables out of $p$ possible, available variables can involve $\binom{p}{s}$ comparisons, which is computationally challenging even for modest values of $p$ and $s$.

In this study, we develop a new approach for incorporating prior information in an inverse model using predictor variables, while simultaneously selecting the relevant predictor variables for the estimation of the unknown quantity of interest. Following a Bayesian approach, we focus on efficiently computing the maximum a posteriori (MAP) estimate of the posterior distribution for the unknown quantity of interest as well as the predictor coefficients ($\boldsymbol{\beta}$). Note that this algorithm can also be applied to spatial interpolation problems where there are multiple predictor variables (i.e., universal kriging), and we describe the implementation for kriging problems where applicable throughout the manuscript. Overall, this approach has three main contributions:





- We develop a more comprehensive statistical model, a joint model for the unknown quantity of interest ($s$) and the predictor coefficients ($\boldsymbol{\beta}$), with sparsity promoting priors on the coefficients $\boldsymbol{\beta}$ that enables model selection;

- We adapt an iterative algorithm developed by the authors in (Chung et al., 2023) to simultaneously estimate both $s$ and $\boldsymbol{\beta}$. This algorithm requires only a "single-step" to select a set of predictors and solve the inverse problem (i.e., compute estimates for $s$ and $\boldsymbol{\beta}$). The proposed algorithm also automatically selects the regularization parameters on-the-fly;

- We evaluate this algorithm using several examples from AIM, including examples drawn from NASA's Orbiting Carbon Observatory 2 (OCO-2) satellite and compare against existing model selection techniques, on small and moderate sized problems. For the largest problem we consider, existing model selection techniques cannot be run in a reasonable amount
of time.

The paper is organized as follows. In Section 2 we describe previously used strategies for model selection in the context of inverse problems, highlighting their different uses and pitfalls. In Section 3 we present the new proposed strategy, providing a detailed explanation of the hierarchical Bayesian model used for the problem defined in Eqs. (2) and (1), as well as a description of the optimization strategy used to compute the MAP estimator and the subsequent algorithm. Numerical results are provided
in Section 4, and conclusions and future work are described in Section 5.

## 2 Existing strategies for model selection in inverse problems

There are different strategies to decide which predictor variables or prior information to include within this inverse (or kriging) model, operating under the assumptions described in Eqs. (1) and (2). The first approach is to choose predictor variables or prior information based on expert judgment. For example, perhaps a modeler trusts one biogeochemical $CO_2$ or $CH_4$ model
more than others – based either on previous analysis or on existing literature. A downside of this approach is that it often necessitates making subjective decisions that are not necessarily based on the available atmospheric $CO_2$ or $CH_4$ observations.

The second approach used in several inverse studies is model selection. This class of methods is also frequently used in regression analysis and linear modeling (e.g., Ramsey and Schafer, 2013). This approach requires a two-step process. The first step is to run model selection to decide on a set of predictor variables, and the second step is to incorporate those variables
into the inverse model and estimate the unknown quantity ($s$). Usually, the goal is to identify a small set of predictor variables that have the greatest power to predict the observations ($z$). Specifically, consider a linear inverse problem of the form (e.g., Brasseur and Jacob, 2017),

$$z = \mathbf{H}s + \epsilon \tag{2}$$

where $z \in \mathbb{R}^m$ is a vector of observations, so that the variable $m$ indicates the total number of available observations. Here,
$\mathbf{H} \in \mathbb{R}^{m \times n}$ represents a physical model that relates the unknown quantity to the observations, and $\epsilon \in \mathbb{R}^m$ represents noise or errors, including errors in the observations $z$ and in the physical model $\mathbf{H}$. In the present study, as is the prevalent approach, we model this error as normally distributed with zero mean and covariance matrix $\mathbf{R} \in \mathbb{R}^{m \times m}$. Note that for the kriging case, $\mathbf{H}$





contains a single value of 1 in each row; this entry links the observation associated with that row to the column that corresponds to the matching location in the unknown space. All remaining elements of $\mathbf{H}$ are set to $0$. Given these relationships, the goal

of model selection is to find a set of predictor variables ($\mathbf{X}$, Eq. 1) that best improves the fit of $\mathbf{s}$ against available observations $\mathbf{z}$. Specifically, model selection will typically reward combinations of predictor variables that are a better fit against available observations and penalize models for increasing complexity (i.e., for increasing numbers of predictor variables).

Existing model selection methods usually determine this fit using the weighted sum of squares residuals (WSS) (e.g., Kitanidis, 1997; Gourdji et al., 2008):

$$\text{WSS}(\mathcal{S}) = \mathbf{z}^T \left( \mathbf{\Psi}^{-1} - \mathbf{\Psi}^{-1}\mathbf{H}\mathbf{X}_{\mathcal{S}} \left( \mathbf{X}_{\mathcal{S}}^{\mathbf{T}}\mathbf{H}^{\mathbf{T}}\mathbf{\Psi}^{-1}\mathbf{H}\mathbf{X}_{\mathcal{S}} \right)^{-1} \mathbf{X}_{\mathcal{S}}^{\mathbf{T}}\mathbf{H}^{\mathbf{T}}\mathbf{\Psi}^{-1} \right) \mathbf{z}, \tag{3}$$

where where $\mathcal{S}$ indicates a subset of predictor variables from the full list of possible variables. Specifically, $\mathcal{S} \subset \{1, \ldots, p\}$, and $|\mathcal{S}|$ denotes the total number of predictor variables in the subset. For example, if $p = 25$, $\mathcal{S} = \{1, 4, 7\}$ is a subset of $\{1, \ldots, 25\}$ with a total number of variables $|\mathcal{S}| = 3$. Then we define $\mathbf{X}_{\mathcal{S}}$ as a matrix of size $n \times |\mathcal{S}|$ which contains columns from $\mathbf{X}$ corresponding to the set $\mathcal{S}$. Furthermore, $\mathbf{\Psi} = \mathbf{H}\mathbf{Q}\mathbf{H}^{\mathbf{T}} + \mathbf{R}$ and $\mathbf{Q}$ is a covariance matrix defined by the modeler that describes

the spatial and/or temporal properties of the stochastic component ($\boldsymbol{\zeta}$) in Eq. (1). In writing expressions such as Eq. (3), we assume that $\mathbf{\Psi}$ is positive definite and $\mathbf{H}\mathbf{X}_{\mathcal{S}}$ has full column rank.

Common model selection methods include the partial F-test (a.k.a., the variance ratio test), the Akaike Information Criterion (AIC), and the Bayesian Information Criterion (BIC). The partial F-test is a frequentist statistical test. If the calculated p-value is below a certain threshold (typically a p-value $< 0.05$), then it is advisable to add or keep a specific predictor variable within

the model (e.g., Ramsey and Schafer, 2013). This test can only be used to compare two candidate models at a time, a model with a smaller number of predictor variables, $\mathcal{S}$ with $|\mathcal{S}| = s$, and a model with a larger number of predictor variables, $\mathcal{T}$ with $|\mathcal{T}| = s + t$. The partial F-test further entails calculating the WSS for $\mathcal{S}$ and $\mathcal{T}$ to quantify how well each model matches available observations as in (3). The improvement in model fit is evaluated using

$$\nu(\mathcal{S}, \mathcal{T}) = \frac{(\text{WSS}(\mathcal{S}) - \text{WSS}(\mathcal{T}))/t}{\text{WSS}(\mathcal{T})/(p - (s + t))}, \tag{4}$$

where the level of significance is quantified using an $F$ distribution with $s$ and $p - (s + t)$ degrees of freedom. A small p-value indicates that $\text{WSS}(\mathcal{T})$ and $\text{WSS}(\mathcal{S})$ are significantly different and the larger model $\mathcal{T}$ is preferable to the smaller model $\mathcal{S}$. To implement the partial F-test, a modeler usually needs to compare many different combinations for $\mathcal{S}$ and $\mathcal{T}$ to decide on the best overall model (e.g., Ramsey and Schafer, 2013).

The AIC and BIC, by contrast, operate on a different principle (Bozdogan, 1987; Schwarz, 1978). A modeler will often

calculate an AIC or BIC score for every possible combination of predictor variables (Gourdji et al., 2012; Miller et al., 2013):

$$\text{AIC}(\mathcal{S}) = \ln|\mathbf{\Psi}| + \text{WSS}(\mathcal{S}) + |\mathcal{S}| \tag{5}$$

$$\text{BIC}(\mathcal{S}) = \ln|\mathbf{\Psi}| + \text{WSS}(\mathcal{S}) + |\mathcal{S}|\ln(m) \tag{6}$$

where $\mathcal{S}$ is a subset of $\{1, \ldots, p\}$, $\ln$ refers to the natural logarithm (with base $e$), and $|\mathbf{\Psi}|$ denotes the determinant of the matrix $\mathbf{\Psi}$. The combination of predictor variables with the lowest AIC or BIC score is deemed the best model. Note that these two





approaches are conceptually similar but have different penalty terms for model complexity: the penalty in the BIC depends on the number of observations ($m$) while the AIC penalty does not.

An upside of statistical model selection is objectivity; this approach can be used to choose predictor variables that are best able to reproduce the observations without over-fitting those observations. There are several downsides to this approach. First, it requires multiple steps—the user needs to run model selection to identify the predictor variables and then subsequently solve the inverse problem to estimate the unknowns $\mathbf{s}$. Second, there are often computational compromises required to implement model selection. If there are $p$ possible predictor variables, there are $2^p$ different combinations to evaluate that range in size from 0 to $p$ total predictor variables. One possible way to reduce the size of this search space is to implement the partial F-test using forward, backward, or stepwise selection (e.g., Ramsey and Schafer, 2013; Gourdji et al., 2008). In forward selection, one would start without any predictor variables in the model and progressively try to add more predictor variables. In backward selection, one would start with all predictor variables and progressively try to remove individual variables from the model (i.e., progressively remove variables for which the p-value $> 0.05$). Stepwise selection, by contrast, alternates between forward and backward selection at each iteration. These strategies reduce the number of candidate model combinations to evaluate, but these three strategies are not guaranteed to converge on the same final result. Beyond the partial F-test, existing studies have also laid out strategies to narrow the number of combinations that need to be evaluated when implementing the AIC or BIC; these strategies, known as branch and bound algorithms, attempt to eliminate multiple, related combinations or branches with each calculated BIC score (e.g., Yadav et al., 2013).

Third, this model selection may not work at all for very large inverse problems due to computational limitations. For example, existing inverse modeling and kriging studies that implement model selection do so by calculating WSS for different combinations of predictor variables as defined in Eq. (3). This equation requires formulating and inverting $\boldsymbol{\Psi}$, a task that is not computationally feasible for many large inverse problems or for problems where $\mathbf{H}$ is not explicitly available as a matrix but rather where only the outputs of forward and adjoint models are available. More recently, a handful of studies have replaced $\boldsymbol{\Psi}^{-1}$ with an approximate, diagonal matrix (Miller et al., 2018, 2020b; Chen et al., 2021a, b; Zhang et al., 2023). This compromise makes it possible to estimate WSS, but a downside is that this approximation for $\boldsymbol{\Psi}^{-1}$ does not match the actual values of $\mathbf{R}$ and $\mathbf{Q}$ that are used in solving the inverse problem.

## 3 Proposed Approach: msHyBR

In this study, we develop a new approach `msHyBR` for incorporating prior information or predictor variables into an inverse model. This approach directly addresses several of the challenges described above. First, it is computationally efficient, even for very large inverse problems with many observations and/or unknowns where it is not possible to compute $\boldsymbol{\Psi}^{-1}$ (or more appropriately, solve linear systems with $\boldsymbol{\Psi}$) and for problems where an explicit $\mathbf{H}$ matrix is not available. Second, the computational cost of this approach does not scale exponentially with the number of predictor variables $p$ (in contrast with other methods that scale with the number of combinations, i.e., $2^p$). Third, the proposed approach will determine a set of predictor variables and solve for the unknown quantity $\mathbf{s}$ in a single step, as opposed to the two-step process required for existing model selection





algorithms. Overall, the proposed approach opens the door for assimilating large amounts of prior information or predictor variables within an inverse model. We argue that this capability is important, particularly as the number of environmental,
Earth science, and remote sensing data sets continues to grow.

### 3.1   Model structure and assumptions

A key aspect of this work is proposing a more comprehensive statistical modeling of the problem defined by Eqs. (1) and (2). Recall that $\mathbf{s}$ is the unknown quantity of interest in the inverse problem in Eq. (2), and several predictor variables are used in the estimation of this quantity, as stated in Eq. (1). In this work, we model the coefficients of the predictors ($\boldsymbol{\beta}$) as
random instead of deterministic variables, a contrast to many existing inverse modeling studies (e.g., Kitanidis and VoMvoris, 1983; Michalak et al., 2004). Previous works assume $p \ll n$ and include standard assumptions for coefficients in $\boldsymbol{\beta}$ such as an improper hyperprior Miller et al. (2020a); Saibaba and Kitanidis (2015) or a Gaussian distribution Cho et al. (2022). However, there are many scenarios where $p$ may be large, and we propose a new model suitable for these cases, imposing a sparsity-promoting prior on $\boldsymbol{\beta}$. The sparsity, in turn, determines which entries of $\boldsymbol{\beta}$ (and corresponding predictor variables or columns
of $\mathbf{X}$) are meaningful. The proposed model has the following hierarchical form:

$$\mathbf{s} = \mathbf{X}\boldsymbol{\beta} + \boldsymbol{\zeta}, \quad \boldsymbol{\zeta} \sim \mathcal{N}(\mathbf{0}, \lambda^{-2}\mathbf{Q}), \quad \beta_j \sim \mathrm{Laplace}(0, 2\alpha^{-2}) \quad \text{for } 1 \le j \le p, \tag{7}$$

where the predictor coefficients ($\boldsymbol{\beta}$) follow a Laplace distribution that promotes sparsity, and the parameter $\alpha$ controls the shape of that distribution. Note that we also include a regularization parameter ($\lambda$) that scales the covariance matrix ($\mathbf{Q}$) such that the overall inverse model is consistent with the actual model-observation residuals. Finally, note that the overall model structure
with this regularization parameter can be reformulated more compactly as follows:

$$\mathbf{s} \mid \boldsymbol{\beta} \sim \mathcal{N}(\mathbf{X}\boldsymbol{\beta}, \lambda^{-2}\mathbf{Q}). \tag{8}$$

Assuming (2) and (7) and by Bayes' theorem, the density function of the joint posterior probability of $\mathbf{s}$ and $\boldsymbol{\beta}$ is

$$\begin{aligned} \pi_{\mathrm{post}}(\mathbf{s}, \boldsymbol{\beta} \mid \mathbf{z}) &\propto \pi(\mathbf{z} \mid \mathbf{s}, \boldsymbol{\beta})\pi(\mathbf{s} \mid \boldsymbol{\beta})\pi(\boldsymbol{\beta}) \\ &\propto \exp\left(-\frac{1}{2}\|\mathbf{z} - \mathbf{Hs}\|_{\mathbf{R}^{-1}}^2 - \frac{\lambda^2}{2}\|\mathbf{s} - \mathbf{X}\boldsymbol{\beta}\|_{\mathbf{Q}^{-1}}^2 - \frac{\alpha^2}{2}\|\boldsymbol{\beta}\|_1\right). \end{aligned} \tag{9}$$

Here, all terms in the exponent are vector norms and, in particular, $\|\mathbf{x}\|_{\mathbf{L}}^2 = \mathbf{x}^T\mathbf{L}\mathbf{x}$ for any symmetric positive definite (SPD)
matrix $\mathbf{L}$. Further, the symbol $\propto$ denotes proportionality. Note that the joint posterior probability for $\mathbf{s}$ and $\boldsymbol{\beta}$ is not Gaussian; however, the MAP estimate can be computed by solving the following optimization problem,

$$\min_{\mathbf{s}, \boldsymbol{\beta}} \left\{ \frac{1}{2}\|\mathbf{z} - \mathbf{Hs}\|_{\mathbf{R}^{-1}}^2 + \frac{\lambda^2}{2}\|\mathbf{s} - \mathbf{X}\boldsymbol{\beta}\|_{\mathbf{Q}^{-1}}^2 + \frac{\alpha^2}{2}\|\boldsymbol{\beta}\|_1 \right\}. \tag{10}$$

Other Bayesian models can be used for sparsity promotion, see e.g., Calvetti et al. (2020).

The remainder of this section is dedicated to describing a computationally efficient algorithm called msHyBR to solve (10),
i.e., to estimate $\mathbf{s}$ and $\boldsymbol{\beta}$ simultaneously. msHyBR is an iterative procedure that takes as input both the problem inputs and





the set of predictor variables in $\mathbf{X}$. At each iteration, a projected problem is solved and the solution subspace is expanded, until some stopping criteria is satisfied. Reconstructed estimates of $\mathbf{s}$ and $\boldsymbol{\beta}$ are the outputs of the algorithm, and subsequent thresholding of $\boldsymbol{\beta}$ can be done, e.g., to identify important predictor variables. A general overview of the approach is provided in Figure 1.

## 3.2 Methodology and algorithm

To handle the computational burden of computing the inverse or square root of the covariance matrix ($\mathbf{Q}$), we begin by transforming the inverse problem in (10), following Chung et al. (2023); Chung and Saibaba (2017). This transformation is crucial for many high dimensional problems where $\mathbf{Q}$ can only be accessed through matrix-vector products. Let

$$\boldsymbol{\gamma} = \mathbf{Q}^{-1}(\mathbf{s} - \mathbf{X}\boldsymbol{\beta}), \tag{11}$$

so that the solution of the problem defined in (10) can subsequently be obtained by solving the following optimization problem:

$$\min_{\boldsymbol{\gamma},\boldsymbol{\beta}} \left\{ \frac{1}{2} \left\| \mathbf{z} - \begin{bmatrix} \mathbf{HQ} & \mathbf{HX} \end{bmatrix} \begin{bmatrix} \boldsymbol{\gamma} \\ \boldsymbol{\beta} \end{bmatrix} \right\|_{\mathbf{R}^{-1}}^2 + \frac{\lambda^2}{2} \|\boldsymbol{\gamma}\|_{\mathbf{Q}}^2 + \frac{\alpha^2}{2} \|\boldsymbol{\beta}\|_1 \right\}. \tag{12}$$

For problems where it is assumed that the predictor coefficients $\boldsymbol{\beta}$ follow a Gaussian distribution, generalized hybrid projection methods were described in Cho et al. (2022). However, solving (12) is more difficult due to the $\ell_1$-regularizer, which is not differentiable at the origin. Several techniques have been devised to approximate the solution of similar inverse problems. One approach is to use nonlinear optimization methods, particularly iterative shrinkage algorithms such as FISTA Beck and Teboulle (2009), or separable approximations such as SPARSA Wright et al. (2008).

Alternately, one can use a majorization-minimization (MM) approach, which involves successively minimizing a sequence of quadratic tangent majorants of the original functional centered at each approximation of the solution. For example, methods based on iterative schemes that approximate the $\ell_1$-norm regularization term by a sequence of weighted $\ell_2$ terms, and in combination with Krylov methods, are usually referred to as iterative re-weighted norm (IRN) schemes Rodríguez and Wohlberg (2008); Daubechies et al. (2010). In this paper, we build on this strategy. In particular, for Eq. (12), and given an initial guess $(\boldsymbol{\gamma}^{(0)}, \boldsymbol{\beta}^{(0)})$, we can solve a sequence of reweighted least squares problems of the form

$$(\boldsymbol{\gamma}^{(k+1)}, \boldsymbol{\beta}^{(k+1)}) = \underset{\boldsymbol{\gamma},\boldsymbol{\beta}}{\arg\min} \left\{ \frac{1}{2} \left\| \mathbf{z} - \begin{bmatrix} \mathbf{HQ} & \mathbf{HX} \end{bmatrix} \begin{bmatrix} \boldsymbol{\gamma} \\ \boldsymbol{\beta} \end{bmatrix} \right\|_{\mathbf{R}^{-1}}^2 + \frac{\lambda^2}{2} \|\boldsymbol{\gamma}\|_{\mathbf{Q}}^2 + \frac{\alpha^2}{2} \|\mathbf{D}(\boldsymbol{\beta}^{(k)})\boldsymbol{\beta}\|_2^2 \right\}, \tag{13}$$

where $\mathbf{D}(\boldsymbol{\beta})$ is an invertible diagonal matrix constructed such that:

$$\mathbf{D}(\boldsymbol{\beta}) = \mathrm{diag}\left( \left[ 2\sqrt{\beta_i^2 + \epsilon} \right]^{-1/2} \right)_{i=1}^p \quad \text{and} \quad \|\boldsymbol{\beta}\|_1 \leq \|\mathbf{D}(\boldsymbol{\beta}^{(k)})\boldsymbol{\beta}\|_2^2 + C$$

with a positive constant $C$ independent of $\boldsymbol{\beta}$. Then, each step of the MM approach consists of solving problem (13). For high dimensional problems, it is unfeasible to solve (13) directly, but an iterative method such as the generalized hybrid method





**Figure 1.** General approach for simultaneous model selection and inversion. Given both problem inputs and predictor variables, msHyBR is an iterative procedure that solves a projected problem (with automatic estimation of parameters $\lambda$ and $\alpha$) and expands the solution subspace, until some stopping criteria is satisfied. Reconstructed estimates of $\mathbf{s}$ and $\boldsymbol{\beta}$ can be used for further analysis, i.e., to identify important predictor variables $\mathbf{X}_{\mathcal{S}}$, where $\mathcal{S}$ is the set of selected indices.





described in Cho et al. (2022) could be used. However, this strategy has two main disadvantages: (1) using an iterative method yields an inner-outer optimization scheme, which can be very computationally expensive; (2) the regularization parameters $\lambda$ and $\alpha$ in (13) cannot be selected independently since previous algorithms require assuming that $\alpha = \tau\lambda$ for some known $\tau > 0$ (even if $\alpha$ can be computed automatically).

To overcome these shortcomings, we use flexible Krylov methods, which use iteration-dependent preconditioning to build a suitable basis for the solution and have been shown to be very competitive to solve problems involving $\ell_1$-norm regularization in other contexts (e.g., see Chung and Gazzola, 2019; Gazzola et al., 2021). In particular, flexible Krylov methods are iterative hybrid projection schemes and so they are characterized by two main components: the (single) solution subspace that is generated and the optimality conditions that are imposed to compute an approximate solution at each iteration.

Note that since we are considering a joint model, we need to build a solution space for both the variable $\boldsymbol{\gamma}$, which is related to the unknown quantity of interest (s) through the efficient change of variables defined in Eq. (11), and the predictor coefficients ($\boldsymbol{\beta}$). Leveraging flexible Krylov methods in a similar fashion to Chung et al. (2023), we use the Flexible Generalized Golub-Kahan (FGGK) process to generate a basis for the solution, which is augmented with a new basis vector at each iteration.

### 3.2.1 Flexible Generalized Golub-Kahan (FGGK) iterative process.

The FGGK process is an iterative procedure that constructs a basis for $[\boldsymbol{\gamma}^T, \boldsymbol{\beta}^T]^T$, and where each basis vector is stored as columns of $\mathbf{Z}_k$. First, the initialization step consists of computing $m_{1,1} = \|\mathbf{z}\|_{\mathbf{R}^{-1}}$, $\mathbf{u}_1 = \mathbf{z}/m_{1,1}$, $\mathbf{v}_1 = \mathbf{H}^\top\mathbf{R}^{-1}\mathbf{u}_1$ and $t_{1,1} = \|\mathbf{v}_1\|_{\mathbf{Q}}$. At each iteration $k$, the FGGK process generates vectors $\mathbf{z}_k$, $\mathbf{v}_k$, and $\mathbf{u}_{k+1}$ by updating the following relation:

$$m_{k+1,k}\mathbf{u}_{k+1} = \mathbf{HQv}_k + \mathbf{HXD}_k^{-1}\mathbf{X}^T\mathbf{v}_k - \sum_{j=1}^{k} m_{j,k}\mathbf{u}_j \tag{14}$$

$$t_{k+1,k}\mathbf{v}_{k+1} = \mathbf{H}^\top\mathbf{R}^{-1}\mathbf{u}_{k+1} - \sum_{j=1}^{k} t_{j,k}\mathbf{v}_j. \tag{15}$$

Generally, this process involves constructing new direction vectors and orthonormalizing them using appropriate inner-products. In particular, $\mathbf{u}_{k+1}$ is constructed considering $\mathbf{u} = (\mathbf{HQ} + \mathbf{HD}_k^{-1}\mathbf{X}^T)\mathbf{v}_k$, orthogonalizing $\mathbf{u}$ against the previous basis vectors $\mathbf{u}_1, \ldots, \mathbf{u}_k$ using the inner product defined by $\mathbf{Q}$, and normalizing this using the corresponding norm induced by this inner product. The analogous process is used to construct $\mathbf{v}_{k+1}$, where $\mathbf{v} = \mathbf{H}^\top\mathbf{R}^{-1}\mathbf{u}_{k+1}$ is orthonormalized with respect to the previous vectors $\mathbf{v}_1, \ldots, \mathbf{v}_k$ using the inner product defined by $\mathbf{R}^{-1}$.

Equivalently, we can consider the matrices $\mathbf{U}_{k+1} = \begin{bmatrix} \mathbf{u}_1 & \ldots & \mathbf{u}_{k+1} \end{bmatrix} \in \mathbb{R}^{m \times (k+1)}$ and $\mathbf{V}_{k+1} = \begin{bmatrix} \mathbf{v}_1 & \ldots & \mathbf{v}_{k+1} \end{bmatrix} \in \mathbb{R}^{n \times (k+1)}$ so that, by construction,

$$\mathbf{U}_{k+1}^T\mathbf{R}^{-1}\mathbf{U}_{k+1} = \mathbf{I}_{k+1} \qquad \mathbf{V}_{k+1}^T\mathbf{Q}\mathbf{V}_{k+1} = \mathbf{I}_{k+1}, \tag{16}$$

in exact arithmetic. Moreover, one can define the following augmented matrices

$$\widehat{\mathbf{H}} = \begin{bmatrix} \mathbf{H} & \mathbf{HX} \end{bmatrix} \in \mathbb{R}^{m \times (n+p)} \qquad \text{and} \qquad \widehat{\mathbf{Q}} = \begin{bmatrix} \mathbf{Q} & \\ & \mathbf{I} \end{bmatrix} \in \mathbb{R}^{(n+p) \times (n+p)}, \tag{17}$$





so that Eqs. (14) and (15) can be expressed more compactly as:

$$\widehat{\mathbf{H}}\widehat{\mathbf{Q}}\mathbf{Z}_k = \mathbf{U}_{k+1}\mathbf{M}_k \quad \text{and} \quad \mathbf{H}^T\mathbf{R}^{-1}\mathbf{U}_{k+1} = \mathbf{V}_{k+1}\mathbf{T}_{k+1}. \tag{18}$$

Here, $\mathbf{M}_k \in \mathbb{R}^{(k+1)\times k}$ is upper Hessenberg and $\mathbf{T}_{k+1} \in \mathbb{R}^{(k+1)\times(k+1)}$ is upper triangular, and the solution space for $[\boldsymbol{\gamma}^T, \boldsymbol{\beta}^T]^T$ is spanned by the columns of $\mathbf{Z}_k$, and it is defined as

$$\mathbf{Z}_k = \begin{bmatrix} \mathbf{z}_1 & \dots & \mathbf{z}_k \end{bmatrix} = \begin{bmatrix} \mathbf{v}_1 & \dots & \mathbf{v}_k \\ \mathbf{w}_1 & \dots & \mathbf{w}_k \end{bmatrix} = \begin{bmatrix} \mathbf{V}_k \\ \mathbf{W}_k \end{bmatrix} \in \mathbb{R}^{(n+p)\times k}. \tag{19}$$

### 3.2.2 Computation of the solution.

After a new basis vector is included in the solution space, one needs to (partially) solve a subproblem as defined in (13). That is, at each iteration, we solve a small projected least-squares problem to approximate the solution of problem (13) in a space of increasing dimension. In particular, using the relations in Eqs. (16) and (18) obtained using the FGGK process, at each iteration $k$ we solve

$$\min_{\boldsymbol{\gamma}=\mathbf{V}_k\mathbf{f}, \boldsymbol{\beta}=\mathbf{W}_k\mathbf{f}} \|\mathbf{H}\mathbf{Q}\boldsymbol{\gamma} + \mathbf{H}\mathbf{X}\boldsymbol{\beta} - \mathbf{z}\|_{\mathbf{R}^{-1}}^2 + \lambda^2\|\boldsymbol{\gamma}\|_{\mathbf{Q}}^2 + \alpha^2\|\boldsymbol{\beta}\|_2^2, \tag{20}$$

for $\boldsymbol{\gamma}_k$ and $\boldsymbol{\beta}_k$, or, equivalently, $\boldsymbol{\gamma}_k = \mathbf{V}_k\mathbf{f}_k$ and $\boldsymbol{\beta}_k = \mathbf{W}_k\mathbf{f}_k$ where

$$\mathbf{f}_k = \underset{\mathbf{f}\in\mathbb{R}^k}{\arg\min} \|\mathbf{M}_k\mathbf{f} - m_{1,1}\mathbf{e}_1\|_2^2 + \lambda^2\|\mathbf{f}\|_2^2 + \alpha^2\|\mathbf{W}_k\mathbf{f}\|_2^2. \tag{21}$$

Finally, we need to undo the change of variables defined in Eq. (11), so that the solution of the original problem in Eq. (2) is approximated at each iteration by $\mathbf{s}_k = \mathbf{Q}\boldsymbol{\gamma}_k + \mathbf{X}\boldsymbol{\beta}_k$. Note that, to simplify the notation when deriving the model, we assumed that the regularization parameters $\lambda$ and $\alpha$ are fixed, but these can be automatically updated at each iteration so that, effectively, $\lambda = \lambda_k$ and $\alpha = \alpha_k$ in Eq. (21). This procedure will be explained in the following section. The pseudo-code for the new model selection HyBR method (msHyBR) can be found in Algorithm 1.

Notice that (21) is a standard least-squares problem with two Tikhonov regularization terms, and the coefficient matrix is of size $(k+1)\times k$, so the solution can be computed efficiently Björck (1996). Efficient QR updates for $\mathbf{W}_k$ were considered in Chung et al. (2023).

Each iteration of msHyBR requires one matrix-vector multiplication with $\mathbf{H}$ and its adjoint (let $T_{\mathbf{H}}$ denote the cost one matrix-vector product with $\mathbf{H}$ or its adjoint), two matrix-vector multiplication with $\mathbf{X}$ and one with its adjoint (similarly, denoted as $T_{\mathbf{X}}$), two matrix-vector products with $\mathbf{Q}$ ( denoted as $T_{\mathbf{Q}}$), one matrix-vector product with $\mathbf{R}^{-1}$ (denoted as $T_{\mathbf{R}^{-1}}$), one matrix-vector product with $\mathbf{D}_k^{-1}$ (denoted as $T_{\mathbf{D}_k^{-1}}$), the inversion of a diagonal matrix $\mathbf{D}_k^{-1}$ that is $\mathcal{O}(p)$ floating point operations (flops) and an additional $\mathcal{O}(k(m+n))$ flops for the summation calculation in (14) and (15). To compute the solution of the projected problem (21), the cost is $\mathcal{O}(k^3)$ flops, since $\mathbf{M}_k$ is upper Hessenberg. And the cost of forming $\boldsymbol{\gamma}$ and $\boldsymbol{\beta}$ to obtain $\mathbf{s}_k$ is $\mathcal{O}(k(n+p))$. Since $p, k \ll m$ and $p, k \ll n$, $T_{\mathbf{X}} \ll T_{\mathbf{H}}$ and $T_{\mathbf{D}_k^{-1}} \ll T_{\mathbf{Q}}$. The overall cost of the msHyBR algorithm is





300 $T_{\mathrm{msHyBR}} = 2kT_{\mathbf{H}} + 2kT_{\mathbf{Q}} + kT_{\mathbf{R}^{-1}} + \mathcal{O}(k^2(m+n))$ flops. (22)

It is important to note that the projected problem (21) for `msHyBR` is much cheaper to solve than (13) within each MM iteration due to its optimization over a lower-dimensional space.

---

**Algorithm 1** hybrid method for model selection (`msHyBR`)

---

**Require:** Matrix $\mathbf{H} \in \mathbb{R}^{m \times n}$, positive definite matrices $\mathbf{Q} \in \mathbb{R}^{n \times n}$ and $\mathbf{R} \in \mathbb{R}^{m \times m}$, vector $\mathbf{z} \in \mathbb{R}^m, \mathbf{X} \in \mathbb{R}^{n \times p}$. Invertible matrix $\mathbf{D}_1 = \mathbf{I}_p \in \mathbb{R}^{p \times p}$.

1: Initialize $\mathbf{u}_1 = \mathbf{z}/m_{1,1}$, where $m_{1,1} = \|\mathbf{z}\|_{\mathbf{R}^{-1}}$ and $\mathbf{v}_1 = \mathbf{0}, k = 1$.

2: **while** stopping criteria are not satisfied **do**

3:      $\mathbf{h} = \mathbf{H}^T \mathbf{R}^{-1} \mathbf{u}_k, t_{j,k} = \mathbf{h}^T \mathbf{Q} \mathbf{v}_j$ for $j = 1, \ldots, k-1$

4:      $\mathbf{h} = \mathbf{h} - \sum_{j=1}^{k-1} t_{j,k} \mathbf{v}_j, t_{k,k} = \|\mathbf{h}\|_{\mathbf{Q}}, \mathbf{v}_k = \mathbf{h}/t_{k,k}$

5:      $\mathbf{z}_k = \begin{bmatrix} \mathbf{v}_k \\ \mathbf{w}_k \end{bmatrix}, \mathbf{V}_k = \begin{bmatrix} \mathbf{v}_1 & \ldots & \mathbf{v}_k \end{bmatrix}, \mathbf{W}_k = \begin{bmatrix} \mathbf{w}_1 & \ldots & \mathbf{w}_k \end{bmatrix}$, where $\mathbf{w}_k = \mathbf{D}_k^{-1} \mathbf{X}^\top \mathbf{v}_k$.

6:      $\mathbf{h} = \mathbf{H}(\mathbf{Q}\mathbf{v}_k + \mathbf{X}\mathbf{w}_k), m_{j,k} = \mathbf{h}^T \mathbf{R}^{-1} \mathbf{u}_j$ for $j = 1, \ldots, k$

7:      $\mathbf{h} = \mathbf{h} - \sum_{j=1}^{k} m_{j,k} \mathbf{u}_j, m_{k+1,k} = \|\mathbf{h}\|_{\mathbf{R}^{-1}}, \mathbf{u}_{k+1} = \mathbf{h}/m_{k+1,k}$

8:      Update QR factorization to obtain $\mathbf{W}_{k+1} = \mathbf{Q}_{k+1} \mathbf{R}_{k+1}$

9:      Solve (21) to get $\mathbf{f}_k(\lambda_k, \alpha_k)$ with selected regularization parameters $\lambda_k, \alpha_k$.

10:      $\mathbf{s}_k = \mathbf{X}\mathbf{W}_k\mathbf{f}_k + \mathbf{Q}\mathbf{V}_k\mathbf{f}_k$.

11:      $\mathbf{D}_{k+1} = \mathbf{D}(\mathbf{W}_k\mathbf{f}_k)$

12:      $k = k + 1$

13: **end while**

14: **return** Approximations $\mathbf{s}_k$ and $\boldsymbol{\beta}_k$

---

### 3.2.3 Regularization parameter choice.

One of the benefits of the proposed method is that it conveniently allows us to automatically estimate $\lambda$ and $\alpha$ in Eq. (21)

305 throughout the iterations. This feature is common in hybrid projection methods for a single parameter, and more recently has been extended to several parameters (e.g., Chung et al., 2023). The overall aim is to find a good regularization parameter for each of the projected subproblems defined in Eq. (21), so that, in practice, $\lambda = \lambda_k$ and $\alpha = \alpha_k$. Let us then consider the unknown quantity of interest at each iteration as a function of the regularization parameters, i.e. $\mathbf{s}_k(\lambda_k, \alpha_k)$.

To validate the potential of the method independently of the regularization parameter choice criteria, we consider first the

310 optimal regularization parameters with respect to the residual norm, that is:

$$\{\lambda_k, \alpha_k\} = \arg\min_{\lambda, \alpha} \|\mathbf{s}_k(\lambda, \alpha) - \mathbf{s}\|_2^2. \tag{23}$$





Note that this is of course unfeasible for real problems where the solution is unknown. In that case, one can use many different regularization parameter choice criteria, see, e.g. Kilmer and O'Leary (2001); Bauer and Lukas (2011); Gazzola and Sabaté Landman (2020); Chung and Gazzola (2021). In this paper, we focus on the discrepancy principle (DP), since it is an established criterion. This consists in choosing $\lambda$ and $\alpha$ such that

$$\{\lambda_k, \alpha_k\} = \arg\min_{\lambda,\alpha} |\|\mathbf{H}\mathbf{s}_k(\lambda,\alpha) - \mathbf{z}\|_{\mathbf{R}^{-1}}^2 - \tau_{\mathrm{DP}} m| = \arg\min_{\lambda,\alpha} |\|\mathbf{M}_k \mathbf{f}_k(\lambda,\alpha) - m_{1,1}\mathbf{e}_1\|_2^2 - \tau_{\mathrm{DP}} m|, \tag{24}$$

where $\tau_{\mathrm{DP}}$ is a predetermined parameter.

Moreover, note that other regularization parameter choices can be used seamlessly using an analogous approach; for example, a few standard choices are described in Chung et al. (2023) for the two variable cases.

## 4 Numerical Experiments

In this section, we present three examples to evaluate the proposed approach `msHyBR` for model selection in inverse problems. First, we present a small one-dimensional signal deblurring example to compare the proposed method with existing model selection approaches and other inverse methods that assume different priors. Second, we develop a hypothetical AIM example featuring a real forward model but synthetic data where the mean, and subsequently the predictor variables, are constructed using a zonation model. Third, we present a case study on estimating $CO_2$ sources and sinks (i.e., $CO_2$ fluxes) across North America using a year of synthetic $CO_2$ observations from NASA's OCO-2 satellite. In this final case study, we use the proposed algorithm to predict synthetic $CO_2$ fluxes using numerous meteorological and environmental predictor variables.

We discuss three different methods for model selection:

1. **Bayesian approaches**: This includes the proposed `msHyBR` approach which performs model selection in a one-step manner.

   (a) **genHyBR**, proposed in (Chung and Saibaba, 2017), corresponds to the known mean case (which we assume to be zero).

   (b) **genHyBRmean**, proposed in (Cho et al., 2020, Algorithm B1), which estimates the mean coefficients $\beta$ using a Gaussian prior; that is, no sparsity is imposed. This method requires estimating two parameters $\lambda$ and $\alpha$, but it requires taking $\alpha = \tau\lambda$, where $\tau$ is a fixed parameter for a specific application.

   Strictly speaking, the latter two methods do not perform model selection but we include them for comparison. If the prior precision $\lambda^2$ is estimated using the relative error in the solution we denote this by "opt", if it is estimated using the discrepancy principle, this is denoted "dp".

2. **Exhaustive selection**: We use the AIC and BIC criteria with exhaustive search (denoted as 'exh' in the results).

3. **Forward selection**: Since the variance ratio test is based on a pair-wise comparison, an exhaustive search would require implementing the test $2^{p-1}!$ times, which is infeasible to calculate for large $p$ (e.g, for $p = 7$, we require $64! \approx 10^{89}$





evaluations). Therefore, we perform the variance ratio test with a forward selection method. For comparison, we also include AIC and BIC with forward selection (denoted as 'fwd' in the results).

## 4.1 One-dimensional deblurring example

The first case study discussed here is a simple 1-D inverse modeling example where the solution is a combination of several polynomial functions. We use this hypothetical case study to examine the basic behavior of different model selection algorithms, including the proposed `msHyBR` algorithm and exhaustive combinatorial search for the optimal solution. We apply these algorithms to more complicated and challenging problems in subsequent case studies.

More specifically, this example concerns an application in 1-D signal deblurring, involving a Gaussian blur with blurring

parameter $\sigma = 1$. The elements in the matrix $\mathbf{H} \in \mathbb{R}^{100 \times 100}$ representing the forward model are defined as

$$\mathbf{H}_{ij} = \frac{1}{\sqrt{2\pi\sigma^2}} \exp\left(-\frac{(i-j)^2}{2\sigma^2}\right) \qquad 1 \leq i, j \leq 100.$$

We assume that the covariates involve 7 predictor variables corresponding to the discretized representations of the first 5 Chebyshev polynomial basis functions of the first kind and 2 two mirrored Heavyside functions with a jump at $0.5$ evaluated on a grid with 100 equispaced points between 0 and 1. The columns of $\mathbf{X} \in \mathbb{R}^{100 \times 7}$ are displayed in Figure 2. Moreover, the

covariance matrix $\mathbf{Q}_s \in \mathbb{R}^{100 \times 100}$ is constructed using a Matérn covariance kernel, see e.g. Chung and Saibaba (2017), with parameters $\nu = 0.5$ and $\ell = 1$. For this example, the exact solution has been created using a realization of the model in (7), with the parameter $\lambda^{-2} = 0.01$. The sparse true coefficients $\boldsymbol{\beta} \in \mathbb{R}^7$ are displayed in Figure 4. Last, the covariance of the additive noise is $\mathbf{R} = 0.1^2 \cdot \mathbf{I} \in \mathbb{R}^{100 \times 100}$. The measurements $\mathbf{z} \in \mathbb{R}^{100}$, along with their noiseless counterpart, are displayed in Figure 2. Note that for this specific noise realization, the noise level $\gamma = \|\mathbf{e}\|_2 / \|\mathbf{Hs}\|_2$ is 0.08.

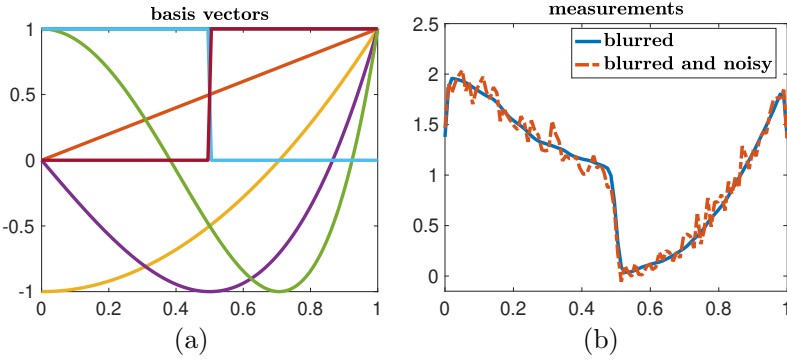

**Figure 2.** For the one-dimensional deblurring example, $p = 7$ predictor variables (or columns of $\mathbf{X}$) and measurements with and without noise are provided.

We use the `msHyBR` approach to compute the reconstructions of $\boldsymbol{s}$ and $\boldsymbol{\beta}$. Note that the estimated coefficients, $\boldsymbol{\beta}$, can be used for model selection by selecting the columns corresponding to the coefficients of $\boldsymbol{\beta}$ whose absolute value is higher than a





chosen threshold. Also note that for the two-stage methods based on model selection, the inverse problem is solved only using the subset of selected covariates $\mathbf{X}_\mathcal{S}$ defined in Section 2. This gives rise to a vector of coefficients $\boldsymbol{\beta}_\mathcal{S}$ which is of smaller or equal dimension to $\boldsymbol{\beta}$, since it corresponds only to the coefficients associated with the previously selected columns. To
compare the estimated and true coefficients, and as observed in Figure 4, the vectors $\boldsymbol{\beta}_\mathcal{S}$ have been augmented with zeroes on the coefficients whose indices are not in the set of selected indices $\mathcal{S}$.

### 4.1.1 Comparison of the reconstructions

For this example, the new method `msHyBR` is competitive in terms of the quality of the reconstructions of $\boldsymbol{s}$ and the coefficients
$\boldsymbol{\beta}$ as displayed in Figures 3 and 4, respectively. The three panels correspond to Bayesian methods (a), Exhaustive selection
methods (b), and Forward selection methods (c). For the comparison with Bayesian models, we observe from Figure 3 (a) that genHyBR produces an oscillatory solution, which can be attributed to the stochastic component ($\boldsymbol{\zeta}$), since the mean $\mathbf{X}\boldsymbol{\beta} = \mathbf{0}$ in Eq. (1). By contrast, the genHyBRmean reconstruction is smoother than the genHyBR reconstruction, which highlights the importance of including adept predictor variables. The genHyBRmean reconstruction is similar but not as accurate as the proposed `msHyBR` algorithm, which is evident in the relative reconstruction error norms that are provided as a function of the
iteration in Figure 5. This improvement can be attributed to `msHyBR`'s superior reconstruction of the coefficients in $\boldsymbol{\beta}$, see Figure 4 (a). Recall that genHyBRmean assumes a Gaussian distribution on $\boldsymbol{\beta}$, and this assumption has a smoothing effect on the computed $\boldsymbol{\beta}$, causing it to deviate from the true $\boldsymbol{\beta}$.

We observe that the `msHyBR` reconstruction of $\boldsymbol{s}$ is similar to those corresponding to the exhaustive selection and the forward selection approaches. However, from the reconstructions of $\boldsymbol{\beta}$ provided in Figures 4(b) and (c), we observe that `msHyBR`
identifies appropriate weights for the 4th and 6th coefficients. Next, we compare the performance of the methods for model selection by including standard quantitative measures for the evaluation of binary classifiers.

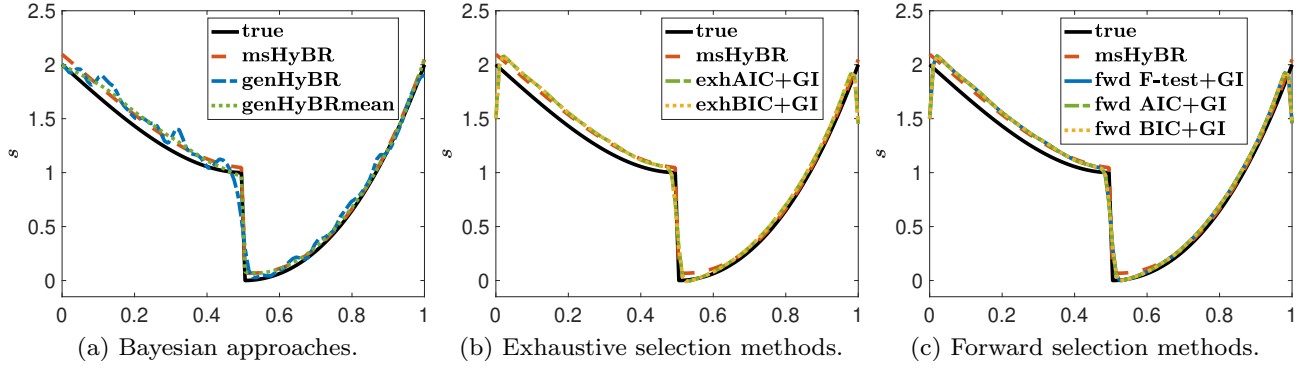

**Figure 3.** Reconstructed quantity of interest (**s**) for the one-dimensional deblurring example with $p = 7$ predictor variables described in Section 4.1 compared to the true solution **s**.





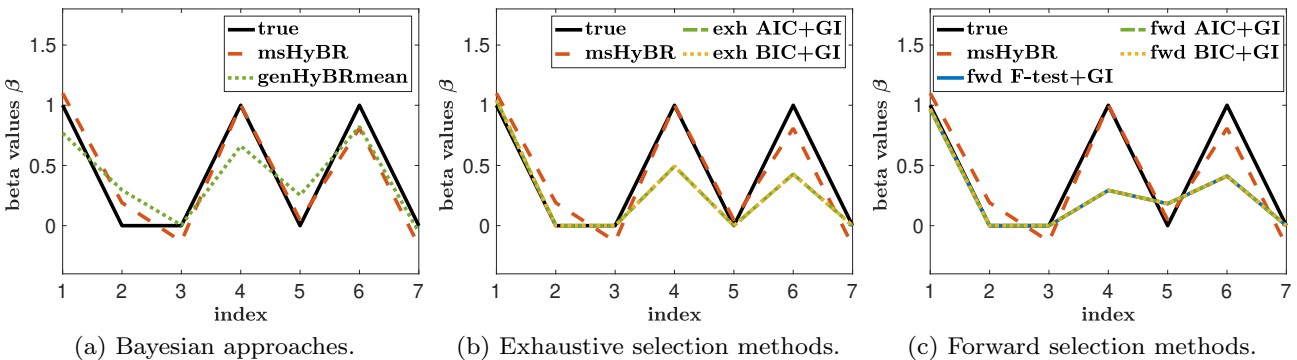

(a) Bayesian approaches.   (b) Exhaustive selection methods.   (c) Forward selection methods.

**Figure 4.** Reconstructed predictor coefficients ($\boldsymbol{\beta}$) for the one-dimensional deblurring example with $p = 7$ predictor variables described in Section 4.1 compared to the true $\boldsymbol{\beta}$.

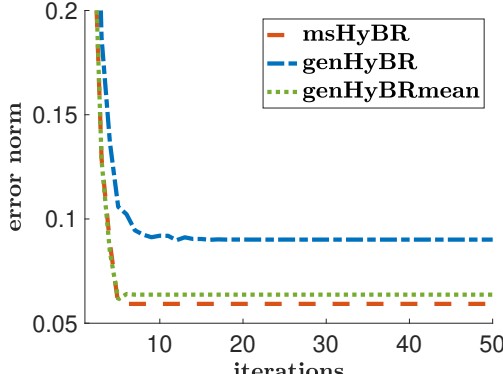

**Figure 5.** Relative error norms history for the one-dimensional deblurring example with $p = 7$ predictor variables described in Section 4.1

### 4.1.2 Comparison of the model selection

We show the different elements in the confusion matrix as well as the F1 score, defined as

$$\text{F1} = \frac{2\text{TP}}{2\text{TP} + \text{FP} + \text{FN}}, \tag{25}$$

where TP and FP are true and false negatives correspondingly and FN are false negatives. Note that F1 can take values between 0 and 1, with 1 corresponding to a perfect classification. Note that the score F1 in (25) corresponds to the harmonic mean of the positive predictive value or precision (fraction of the selected columns that are in the true set) and the sensitivity or recall (fraction of true set of relevant columns that is identified by the method). The results are provided in Table 1, where the best-performing algorithm for each of the categories is highlighted in boldface. For this example, msHyBR performs competitively.

It is also interesting to note that, as expected theoretically, genHyBRmean tends to produce false positives due to the smoothing effect on the coefficients $\boldsymbol{\beta}$.





**Table 1.** Confusion matrix for the one-dimensional deblurring example: true positives (TP), false positives (FP), true negatives (TN), and false negatives (FN). F1 score as defined in (25). The best-performing methods for each category are marked in boldface.

|            | TP | FP | TN | FN | F1   |
|------------|----|----|----|----|------|
| msHyBR     | **3** | **0** | **4** | **0** | **1** |
| exh BIC    | **3** | **0** | **4** | **0** | **1** |
| exh AIC    | **3** | **0** | **4** | **0** | **1** |
| fwd F-test | **3** | 1  | 3  | **0** | 0.86 |
| fwd BIC    | **3** | 1  | 3  | **0** | 0.86 |
| fwd AIC    | **3** | 1  | 3  | **0** | 0.86 |
| genHyBRmean | **3** | 2 | 2  | **0** | 0.75 |

The aim of this example was to evaluate the proposed approach in comparison to existing model selection approaches, for a small problem. Note that all the regularization parameters used to compute the solution of the inverse problem (regardless of whether this is done in one or two steps) are chosen to be optimal with respect to the relative error norm. We consider more

realistic case studies and parameter choice methods in the upcoming sections.

### 4.2 Hypothetical atmospheric inverse modeling (AIM) problem

This experiment concerns a synthetic inverse modeling problem where the true solution features distinct blocks of emissions in different regions of North America (i.e., a zonation model, see Figure 6(a)). This case study is hypothetical, where the ground truth is known. The goal of this case study is to examine the performance of the new msHyBR method compared

to a two-step process where the relevant predictor variables are chosen first (using standard model-selection techniques) and then the solution of the inverse model is computed. The true emissions in this case study have relatively simple geographic patterns, much simpler than most real air pollutant or greenhouse gas emissions. With that said, this case study provides a clear-cut test for evaluating the algorithms developed here; model selection should identify predictor variables (i.e., columns of $\mathbf{X}$) corresponding to the large emissions blocks in the true solution (Figure 6) and should not select predictor variables in

other sub-regions with small or non-existent emissions. Overall, this example demonstrates that msHyBR not only can achieve similar reconstruction quality as a two-step approach, but also can alleviate some of the computational challenges with model selection for larger problems.

In this experiment, we consider a synthetic atmospheric transport problem aimed at estimating the fluxes of an atmospheric tracer across North America, with a spatial resolution of $1° \times 1°$. For this example, we generated synthetic fluxes by only

placing non-zero fluxes in a limited number of regions. These fluxes vary spatially but not temporally. We use a zonation model to build the predictor variables; that is, we divide North America into 78 regions where the inner boundaries correspond to a grid of $10°$ longitude and $7°$ latitude, as can be observed in Figure 6(a). Each column of $\mathbf{X}$ corresponds to an indicator





function for each of the subregions on the grid. Specifically, each column of $\mathbf{X}$ consists of ones and zeros; ones for all model grid boxes that fall within a specific subregion and zero for all other grid boxes. The coefficients $\boldsymbol{\beta}$ determine the weights of each basis vector, and for this example, we use the true values of $\boldsymbol{\beta}$ given in Figure 7(b); the corresponding mean image $\mathbf{X}\boldsymbol{\beta}$ is provided in Figure 6(c). The true emissions in Figure 6(b) were generated as $\mathbf{s} = \mathbf{X}\boldsymbol{\beta} + \boldsymbol{\zeta}$, where $\boldsymbol{\zeta}$ is a realization of $\mathcal{N}(\mathbf{0}, \lambda^{-2}\mathbf{Q})$ with $\mathbf{Q}$ representing a Matérn kernel with parameters $\nu = 2.5$ and $\ell = 0.05$ and $\lambda^{-2} = 0.3$; the same covariance model was used in Chung et al. (2023).

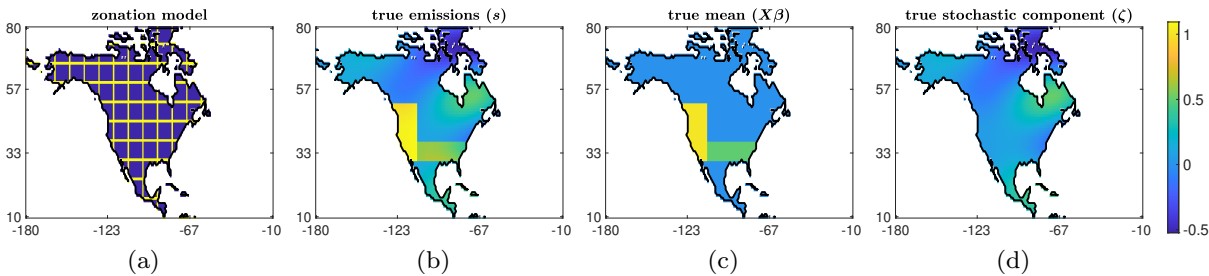

**Figure 6.** Synthetic data used in the atmospheric inverse modeling (AIM) described in Section 4.2. An illustration of the zonation model is provided in (a): the predictor variables (columns of $\mathbf{X}$) correspond to indicator functions in each of the delimited areas evaluated on the grid. The true emissions image in (b) is a sum of the true mean image in (c) and the true stochastic component in (d).

The forward model represented by $\mathbf{H} \in \mathbb{R}^{98880 \times 3222}$ is taken from NOAA's CarbonTracker-Lagrange project Miller et al. (2020a); Liu et al. (2020) and is produced through the Weather Research and Forecasting (WRF) Stochastic Time-Inverted Lagrangian Transport Model (STILT) system Lin et al. (2003); Nehrkorn et al. (2010). The observations $\mathbf{z} \in \mathbb{R}^{98880}$ were simulated based on the spatial and temporal coordinates of OCO-2 observations from July through mid-August 2015 with additive noise following (2), with $\mathbf{R} = 0.183^2 \cdot \mathbf{I}$ corresponding to $\gamma = 0.04$. Given observations in $\mathbf{z}$, forward model $\mathbf{H}$, and predictors $\mathbf{X}$, the goal of AIM is to estimate $\mathbf{s}$.

In a standard two-step approach, the first step is to select a set of important predictors (e.g., columns of $\mathbf{X}$). Exhaustive model selection methods are infeasible for this problem because there are more combinations of predictor variables than we can reasonably compute BIC scores for. Thus, we use a forward selection strategy with the BIC to select the predictor variables. For this example, the set of relevant covariates as determined by forward BIC consisted of the following columns: $\mathcal{S} = \{55, 56, 64, 65, 66, 75\}$. Then with $\mathbf{X}_{\mathcal{S}}$, we solve the inverse problem using genHyBRmean.

The two-step approach has difficulties in regions with a smaller (but still positive) mean. This approach does not select predictor variables in these regions, though these features are broadly captured in the stochastic component. The reconstructions of the coefficients $\beta$ are provided in Figure 7 (b), where we have augmented with zeroes the coefficients whose indices are not in $\mathcal{S}$. Furthermore, the relative reconstruction error norms per iteration of genHyBRmean are provided in Figure 7 (a), and the reconstructions for the two-step approach are provided in the bottom row of Figure 8.

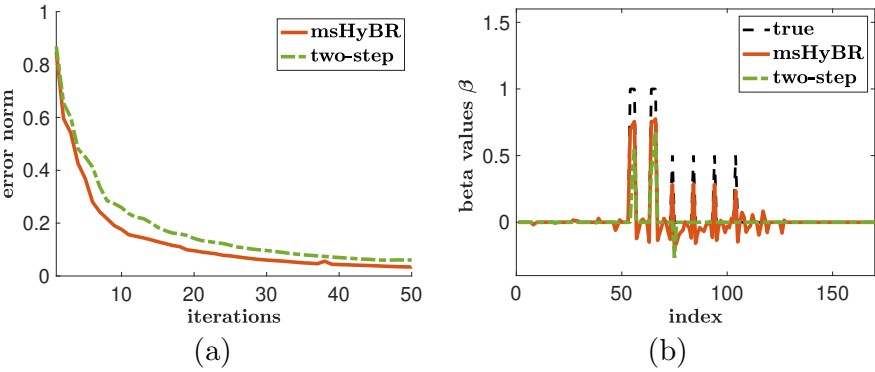

**Figure 7.** (a) Relative reconstruction error norms per iteration of `msHyBR` and the two-step approach (forward BIC + genHyBRmean). (b) Predictor coefficients ($\beta$) for the hypothetical AIM problem.

In contrast to the two-step results described above, the `msHyBR` method selects covariates in all regions that correspond with the true solution. For the `msHyBR` method, we allow the algorithm to simultaneously estimate the predictor variables for the mean, along with the stochastic component. The reconstructions of the emissions **s**, along with the computed mean and stochastic component using `msHyBR` can be found in the top row of Figure 8. The corresponding reconstructions for the two-step approach are provided for comparison. Note that the basis vector representation is constructed via threshholding of the 440   reconstructed $\beta$, which are provided in Figure 7(b). The `msHyBR` method inherently performs model selection by incorporating a sparsity promoting prior on $\beta$, which results in many reconstructed values close to zero. The relative reconstruction error norms per iteration of `msHyBR` provided in Figure 7(a) show similar (and even slightly better) reconstruction quality compared with the two-step approach. Note that the `msHyBR` method achieves a similar reconstruction of the edges surrounding the regions with positive mean emissions compared to the two-step approach.

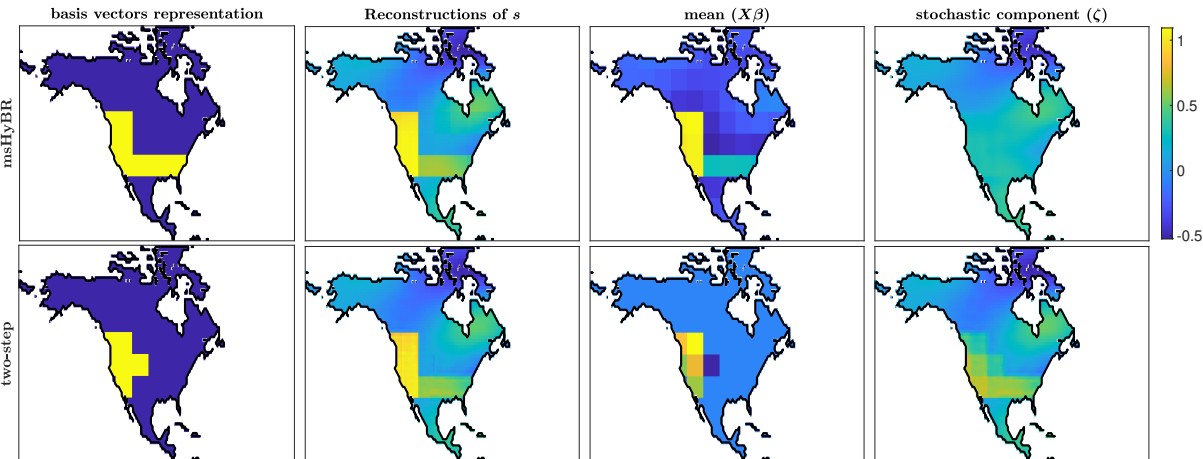

**Figure 8.** Reconstructions for the hypothetical AIM example correspond to `msHyBR` in the top row and a two-step approach in the bottom row. On the left are basis vector representations obtained using selected predictor variables (threshholded for `msHyBR`). Although the reconstructions of **s** in both cases are similar, the forward BIC approach selects fewer predictor variables and hence the stochastic component must resolve the difference. Whereas the `msHyBR` approach simultaneously estimates 78 predictor coefficients (many of which are small) and can estimate a smooth stochastic component (similar to the true images in Figure 9).

For all of the reconstructions, we used the DP to compute the regularization parameter(s), and we take $\mathbf{Q}$ to represent a Matérn kernel with parameters $\nu = 0.5$ and $\ell = 0.5$. We remark that one of the advantages of `msHyBR` is that the covariance scaling factor $\lambda$ in the model, see Eq. (7), can be estimated automatically as part of the reconstruction process. However, for the two-step process, an estimate of $\lambda$ is required for both the model selection process and for reconstruction. For the two-step process, we used $\lambda^{-2} = 0.3$ in the first step since the parameter choice has a big impact on the number of selected covariates, 450    and for the second step, we used the DP within genHyBRmean.

### 4.3    Biospheric CO$_2$ flux example

This AIM case study focuses on CO$_2$ fluxes across North America using synthetic observations based on NASA's OCO-2 satellite. In this case study, the true solution is based on CO$_2$ fluxes in space and time from NOAA's CarbonTracker v2022 product (CT2022) (Jacobson et al., 2023). For illustrative purposes, the true fluxes that have been averaged over time are provided in 455    Figure 9. This product is estimated using in situ CO$_2$ observations and is commonly used across the CO$_2$ community. Unlike the previous hypothetical case study, the fluxes in this example vary every 3 hours (at a $1° \times 1°$ spatial resolution) covering a full calendar year (Sept. 2014 – Aug. 2015). Thus the total number of unknowns for this example is $n = 9.4 \times 10^6$. Like the previous example, forward model simulations are from STILT simulations generated as part of NOAA's CarbonTracker Lagrange project, where the total number of synthetic observations is $m = 9.9 \times 10^4$. We remark that this is a significantly 460    more challenging test case because the problem size is so large that we cannot run comparisons with existing model selection





methods (e.g., AIC or BIC). However, we include this example to highlight the applicability of our approach with very large datasets.

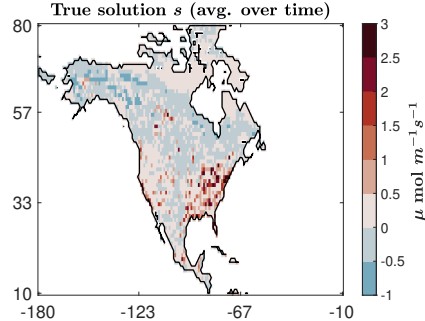

**Figure 9.** True solution representing average CO$_2$ fluxes for the AIM problem based on NASA's OCO-2 satellite observations.

For the predictors of CO$_2$ fluxes, we use a combination of variables, including meteorological variables, in an approach similar to many existing AIM studies of CO$_2$ fluxes (e.g., Gourdji et al., 2008, 2012; Shiga et al., 2018a, b; Randazzo et al.,

2021; Chen et al., 2021a, b; Zhang et al., 2023). The first 12 columns of $\mathbf{X}$ correspond to different spatially constant vectors of ones, each representing one month. The inclusion of these constant columns is common in the AIM studies cited above, and they help account for the fact that CO$_2$ fluxes have a strong seasonal cycle with very different mean fluxes from month to month. Next, we include meteorological variables from the European Centre for Medium-Range Weather Forecasts (ECMWF) Reanalysis v5.1 (ERA-5.1) product (Hersbach et al., 2020) because ERA is also used to help generate CO$_2$ fluxes as part

of the CarbonTracker modeling platform (Jacobson et al., 2023). These 13 variables include temperature at 2 meters above ground, evaporation, mean evaporation rate, mean surface downward short-wave radiation flux, potential evaporation, soil temperature at two different soil levels, total cloud cover, total precipitation, volumetric soil water at two different soil levels, relative humidity, and specific humidity. Several of these variables are highly correlated, and we have included these variables on purpose to examine how the proposed algorithm handles colinearity. In addition to the predictor variables described above,

we include 10 Gaussian random vectors as predictor variables (i.e., as columns of $\mathbf{X}$). These random columns don't have any physical meaning as such but are meant to test the capabilities of inverse modeling algorithm `msHyBR`. Specifically, these vectors should have little to no ability to predict CO$_2$ fluxes, and an interpretable result would be one in which the inverse models either do not select these predictors and/or estimate the corresponding coefficients ($\boldsymbol{\beta}$) to be close to zero.

To produce more realistic scenarios and evaluate the performance of the new algorithm under noisy data, we add randomly

generated Gaussian noise to the synthetic observations. The setup allows us to evaluate how the performance of the proposed inverse modeling approach changes as the noise or error levels change. Note that for a given realization of the noise $\mathbf{e}$, we define the noise level to be $\gamma = \|\mathbf{e}\|_2 / \|\mathbf{Hs}\|_2 = 0.1$ and $0.5$, corresponding to noise levels of 10% (low noise) and 50 % (high noise) respectively. For the covariance matrix $\mathbf{Q}_s$, we use parameters from existing studies that employed this same case study (Miller et al., 2020b; Liu et al., 2020; Cho et al., 2022). Specifically, we use a spherical covariance model with decorrelation

length of 586km and a decorrelation time of 12 days. The diagonal elements of $\mathbf{Q}$ vary by month and have values ranging from





(6.6 ppm)$^2$ to (102 ppm)$^2$ (as in Miller et al., 2020b). Note that within the inverse model, these values are ultimately scaled by the estimated regularization parameter ($\lambda$).

Relative reconstruction error norms per iteration are provided in Figure 10(a). All results correspond to using the DP to select the regularization parameter. We observe that for both noise levels, reconstruction error norms for `msHyBR` follow the
expected behaviour (Figure 7(a)), with slightly smaller errors for the smaller noise level.

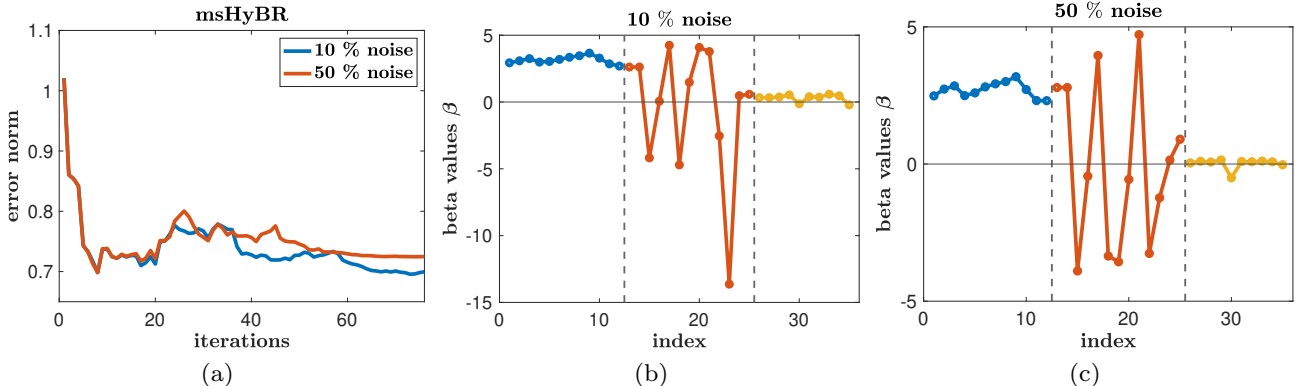

**Figure 10.** Left panel: Relative error norm histories for the AIM problem based on NASA's OCO-2 satellite observations with 10 % and 50 % noise. Reconstructions of the coefficients $\boldsymbol{\beta}$ for 10 % (middle panel) and 50 % (right panel). These results correspond to using the regularization parameters selected by the DP.

The estimated values of $\boldsymbol{\beta}$ using `msHyBR` for the $10\%$ and $50\%$ noise levels are provided in Figures 10(b) and (c) respectively. The dotted lines separate the coefficients corresponding to predictor variables of different natures. The first 12 dots (in blue) represent the seasonality of the mean reconstruction. These estimated coefficients ($\boldsymbol{\beta}$) are roughly constant and bounded away from zero. Interestingly, these coefficients do not describe any seasonal variability in the estimated $CO_2$ fluxes, which was one
of the original motives for allowing the coefficients to vary by season. Rather, the seasonal variability is entirely captured by the meteorological variables and stochastic component. With that said, these coefficients appear to describe a seasonally-averaged mean behavior in the solution.

The second set of coefficients (in red) represents the mixed and potentially collinear set of 13 meteorological variables. We observe that `msHyBR` can identify which of these coefficients are important and which should be dampened (i.e., corresponding
to coefficients close to 0). For different noise levels, we observe that different meteorological variables may be picked up in `msHyBR`. Finally, for the last set of 10 predictor variables that represent spurious and unimportant random vectors, `msHyBR` successfully damps these coefficients at both the noise levels. This demonstrates `msHyBR`'s ability to perform reasonable model selection, via a sparsity promoting prior on $\boldsymbol{\beta}$.

Moreover, the $CO_2$ reconstructions averaged over time can be observed in Figure 11 using `msHyBR` for both noise levels.
Is it interesting to note that, in the low-level scenario, the stochastic component captures more of the variability and therefore





has a strong similarity to the final reconstruction, while in the high noise level case, most of the information of the total reconstruction comes from the mean. At higher noise levels, the inverse model makes less detailed adjustments to the $CO_2$ fluxes via the stochastic component. By contrast, at lower noise levels, the inverse model interprets the observations as being more trustworthy or informative, and the inverse model thus estimates a stochastic component with more spatial variability.

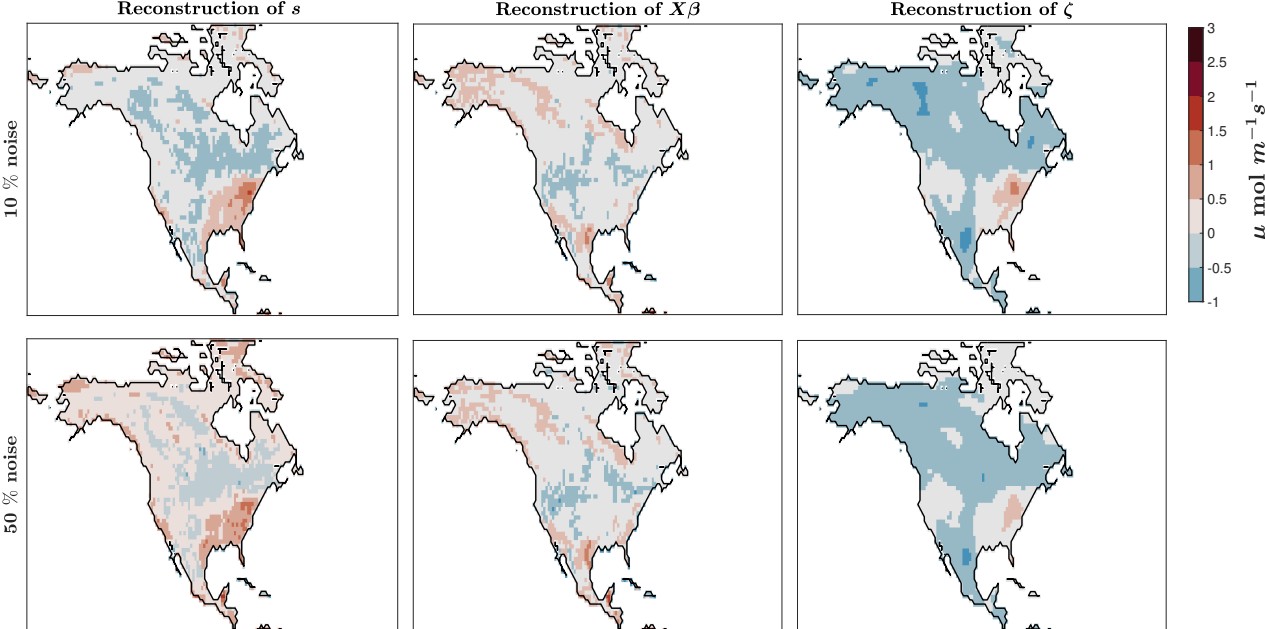

**Figure 11.** Reconstructions of the unknown quantities using the new model selection HyBR for the AIM problem based on NASA's OCO-2 satellite observations in the high noise scenario with 10 % and 50 % noise. Here one can observe the full reconstruction $\mathbf{s}$ (left), the reconstruction of the mean $\mathbf{X}\boldsymbol{\beta}$ (middle), and the reconstruction of the stochastic component $\boldsymbol{\zeta}$ (right). These results correspond to using the regularization parameters selected by the DP.

## 5   Conclusion

This paper presents a set of computationally efficient algorithms for model selection for large-scale inverse problems. Specifically, we describe one-step approaches that use a sparsity-promoting prior for covariate selection and hybrid iterative projection methods for large-scale optimization. A main advantage over existing approaches is that both the reconstruction ($\mathbf{s}$) and the predictor coefficients ($\boldsymbol{\beta}$) can be estimated simultaneously. The proposed iterative approaches can take advantage of efficient matrix-vector multiplications (with the forward model and the prior covariance matrix) and estimate regularization parameters automatically during the inversion process.

Numerical experiments show that the performance of our methods is competitive with widely-used two-step approaches that first identify a small number of important predictor variables and then perform the inversion. The proposed approach is



cheaper (since it avoids expensive evaluations of all possible combinations of candidate predictors), which makes it superior
for problems with many candidate variables (i.e., a large number of columns of $\mathbf{X}$) and limited observations.

Future work includes subsequent UQ and analysis for the hierarchical model (7), which will likely have increased variances
due to additional unknown parameters Cho et al. (2022). Efficient UQ approaches will require MCMC sampling techniques
due to the Laplace assumption. However, it may be possible to use Gaussian approximations and exploit low-rank structure
from the FGGK process, similar to what was done in Cho et al. (2022) for Gaussian distributions.

Overall, we anticipate that algorithms like `msHyBR` will have increasing utility given the plethora of prior information that
is often available for inverse modeling and given the computational need for inverse models that can ingest larger and larger
satellite datasets.

*Code availability.* The MATLAB codes for the one-dimensional deblurring example that were used to generate the results in Sect. 4 are
available at https://zenodo.org/doi/10.5281/zenodo.11164245 (Sabate Landman et al., 2024). Current and future versions of the codes will
also be available at https://github.com/Inverse-Modeling.

*Data availability.* The input files for the case study are available on Zenodo at https://doi.org/10.5281/zenodo.3241466 (Miller et al., 2019).

*Author contributions.* JC, SMM, and AKS designed the study. JJ ran preliminary studies, and MSL conducted the numerical simulations
and comparisons. All the authors contributed to the writing and presentation of results.

*Competing interests.* The contact author has declared that none of the authors has any competing interests.

*Acknowledgements.* This work was partially supported by the National Science Foundation ATD program under grants DMS-2208294,
DMS-2341843, DMS-2026830, and DMS-2026835. CarbonTracker CT2022 results provided by NOAA GML, Boulder, Colorado, USA
from the website at http://carbontracker.noaa.gov. The OCO-2 CarbonTracker-Lagrange footprint library was produced by NOAA/GML and
AER Inc with support from NASA Carbon Monitoring System project Andrews (CMS 2014) Regional Inverse Modeling in North and South
America for the NASA Carbon Monitoring System. We especially thank Arlyn Andrews, Michael Trudeau, and Marikate Mountain for
generating and assisting with the footprint library. Any opinions, findings, conclusions or recommendations expressed in this material are
those of the author(s) and do not necessarily reflect the views of the National Science Foundation.



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
