# Peer review of "A Joint Reconstruction and Model Selection Approach for Large Scale Linear Inverse Modeling (msHyBR v2)"

_Geoscientific Model Development, 2024_

## Referee Comment (RC1)

**Report on "A joint reconstruction and model selection approach for large scale inverse problems"**

Ian Enting

Typeset August 5, 2024.

**Overview**

This paper describes an inversion technique where the regularisation is based by prior predictions from models, with the new feature that selection of an appropriate subset of models is being selected as part of the inversion. It defines an appropriate objective function, and a new algorithm for minimising this objective function.

The technique is then illustrated by three synthetic data examples, firstly a signal processing example and then two inversions of simulated OCO-2 data.

The paper is generally well written, although there are a few points at which a little more detail might make it easier to read. It might benefit from a table of notation — although what appears below was produced for my own benefit while writing this report.

To conclude, this paper is suitable for publication in Geoscientific Model Development. My various comments should be regarded as suggestions for the authors to consider and the editor to take into account, rather than being prescriptive.

**General comments**

- Probably it should be explicitly stated that the formalism as presented here is restricted to linear (or linearised) models.

- As the authors note, model selection (effectively choosing a set of regression variable from a set of $p$ candidate variables) would strictly involve $2^p$ comparisons, and iterative approaches, e.g. successive rejection, still require large numbers of comparisons (and $\binom{p}{q}$ comparisons if $q$, the size of the target subset is pre-determined). The approach here is analogous to the use of $L_1$ norms in regression, fitting a subset of items closely at the expense of downplaying the lack of fit of other items (and thus being less sensitive to outliers).

- Choosing one 'best ' model from a set of models describing the same process seems an appropriate approach. If one has two or more candidate models of different processes that give similar contributions to $\mathbf{z}$, then rather than choose one model, the appropriate conclusion is that the inversion can estimate a linear combination of these models but not distinguish between them. This could be particularly relevant for the third example, where 'lumping' of poorly distinguished weak source regions might be more appropriate than selecting a subset. The example is a good illustration of the power of the method but the method may not be the best way of inverting actual OCO-2 data.

**Line by line**

**83** Suggest using $q$, as above, rather than $s$, as number of target models to avoid confusion with **s**.

**various** Suggest upright T to indicate matrix-vector transpose, since the T is not a mathematical variable. There are various online discussions on best form. My own preference would be $\mathbf{X}^\mathsf{T}$

```
$\mathbf{X}^\mathsf{T}$
```

**various** The importance of this will depend on the fonts used for final production, but in the discussion preprint, the use of bold font to distinguish vectors works poorly for the Greek beta and particularly poorly for gamma. While the intended usage can be deduced from the context, it makes the article a little bit harded to read, since beta does not always appear as a vector (see eqn 7) and both forms are subscripted, once by component index within the vector, and otherwise with subscript to the vector indicating the iteration number.

**259** Strictly, $t_{k+1,k}$ and $m_{k+1,k}$ seem not to be defined at this point. Needs forward reference to Algorithm 1.

**Terms where a few words of description might help**

**Krylov subspace methods** Methods for working with subspaces of large problems..........

**Laplace distribution** Also known as double exponential.

**Notation**

$k$ Iteration count in inversion algorithm. Dimension of the Krylov subspace at that step??

$m$ Number of observations (dimensionality of **z**).

$n$ Number of quantities to be estimated (dimensionality of **s**).

$p$ Number of candidate predictors (models).

**z** Vector of observations that are to be fitted by the inversion.

**s** Vector of quantities to be estimated by the inversion.

$\zeta$ 'Random' component of **s**, assumed to be distributed as zero mean multivariate Gaussian described by **Q**

$\beta$ Vector of contributions from each of the candidate models, where model selection corresponds to taking zero as the estimate of particular components. The selection is 'controlled' by the regularisation parameter $\alpha$.

$\epsilon$ Error component of **z**. Assumed to be zero mean multivariate Gaussian described by **R**.

**R** Covariance matrix for $\epsilon$, the error component of observations, **z** .

**H** Mapping from **s** to observations **z**.

$\lambda$ Regularisation parameter, estimated as part of the invesion, applied as scale factor of **Q** .

$\alpha$ Regularisation parameter, estimated in the inversion, applied Y as defining the scale of the Laplace distribution of $\beta$.

$\gamma$ Transformation such that minimisation over **s**, $\beta$ becomes minimisation over $\gamma$, $\beta$, avoiding the need to invert **Q**.

$\mathbf{u}_k, \mathbf{v}_k$ Additions to the solution subspace after step $k$.

**X** Mapping from candidate models, $\beta$, to **s**.

$j$ Component index within a vector.

---

## Referee Comment (RC2)

**Review of "A Joint Reconstruction and Model Selection Approach for Large Scale Inverse Modeling" (gmd-2024-90) for Geoscientific Model Development**

In this manuscript, the authors propose a modelling and algorithmic approach for performing model selection in large-scale inverse problems. The context is that a set of predictor variables (say, meteorological variables) are used in a linear model to predict an unknown quantity $\mathbf{s}$ (say, surface fluxes). The unknown quantity $\mathbf{s}$ is related to the observations through a forward model. Interest is on both estimating $\mathbf{s}$ and on identifying a small subset of the predictor variables that are most informative of $\mathbf{s}$. To achieve this, the authors propose a sparsity-promoting prior on the coefficients of the predictors, and develop a hybrid iterative projection method based on flexible Krylov subspace methods for efficient optimization.

The paper is well written and the novelty and significance of the work is well justified. The paper is well organized and the figures are clear and informative. The authors perform extensive simulation studies to validate their method. My comments are mostly minor and relate to the (lack of) discussion of prior work in the statistical literature on model selection, as well as some minor points regarding the simulation studies.

My detailed comments are as follows:

1. The proposed model structure in (7) places a Laplace prior on $\boldsymbol{\beta}$, the coefficients of the predictors. This is a common choice for promoting sparsity in the coefficients, which is often called the Bayesian LASSO (Park and Casella, 2008). It would be worthwhile for the authors to discuss this connection: indeed this present manuscript could be seen as the extension of the Bayesian LASSO into the inversion context.

   Park and Casella (2008) also address the problem of performing Markov chain Monte Carlo for this model so it may also be worth mentioning in the Conclusion where uncertainty quantification is discussed.

2. Relatedly, there is literature discussing the implications of the choice of sparsity promoting prior. For example, Carvalho et al. (2010), in proposing an alternative sparsity promoting prior, discuss the limitations of the Laplace prior and other choices, and Piironen and Vehtari (2017) discuss ways to tune sparsity promoting priors. While an extensive discussion of these choices is beyond the scope of this paper, a brief mention of how these modelling choices can impact the results would be useful.

3. I was a bit confused by the one-dimension simulation example in Section 4.1. Here are my questions, which I suggest the authors clarify in the manuscript:

   (a) In Figures 2 and 3, what is the $x$-axis showing?

   (b) In Figure 2(b), can the authors show $\mathbf{s}$ and $\mathbf{X}\boldsymbol{\beta}$ as well as the blurred observation?

   (c) In Figure 3(a), I was surprised to see that all methods noticeably overestimate the true function, particularly for the $x$-axis values between 0 and 0.5. Shouldn't the stochastic component, $\boldsymbol{\zeta}$, correct this? How does this error arise?

   (d) What exactly is the "reconstruction error norm" and the "relative reconstruction error norm" shown in Figure 5 and mentioned in the text (also in the later simulation studies)?

4. In discussing the partial $F$-test in Section 2, it may be worth mentioning that the "smaller" model must nest within the "larger" model, a notable limitation of the $F$-test compared to competing methods.

5. Minor points:

(a) Lines 419–420: the references Miller et al and Liu et al would be better parenthesised.

(b) Lines 485–486: I'm surprised that the elements of $\mathbf{Q}$ have units $(\text{ppm})^2$—isn't this quantity a flux?

**References**

Carvalho, C. M., Polson, N. G., and Scott, J. G. (2010). The horseshoe estimator for sparse signals. *Biometrika*, 97(2):465–480.

Park, T. and Casella, G. (2008). The Bayesian Lasso. *Journal of the American Statistical Association*, 103(482):681–686.

Piironen, J. and Vehtari, A. (2017). Sparsity information and regularization in the horseshoe and other shrinkage priors. *Electronic Journal of Statistics*, 11(2):5018–5051.

---

## Author Comment (AC1)

**Geoscientific Model Development**
**Reply to Referee 1 (Ian Enting)**

"A Joint Reconstruction and Model Selection Approach for Large Scale Inverse Modeling"
by Malena Sabate Landman, Julianne Chung, Jiahua Jiang, Arvind K. Saibaba, and Scot M. Miller

September 25, 2024

We are grateful to Professor Enting for his careful reading and thoughts on our manuscript. Below, we repeat his remarks and interleave our responses. All modifications in the revised manuscript are highlighted in **blue** and all references to equations, pages, lines, and citations correspond to the revised manuscript.

**Overview** This paper describes an inversion technique where the regularisation is based by prior predictions from models, with the new feature that selection of an appropriate subset of models is being selected as part of the inversion. It defines an appropriate objective function, and a new algorithm for minimising this objective function.

The technique is then illustrated by three synthetic data examples, firstly a signal processing example and then two inversions of simulated OCO-2 data.

The paper is generally well written, although there are a few points at which a little more detail might make it easier to read. It might benefit from a table of notation — although what appears below was produced for my own benefit while writing this report.

To conclude, this paper is suitable for publication in Geoscientific Model Development. My various comments should be regarded as suggestions for the authors to consider and the editor to take into account, rather than being prescriptive.

**General comments**

- Probably it should be explicitly stated that the formalism as presented here is restricted to linear (or linearised) models.

  We have included "for linear inverse modeling" in the abstract and "Linear" in the title.

- As the authors note, model selection (effectively choosing a set of regression variable from a set of p candidate variables) would strictly involve $2^p$ comparisons, and iterative approaches, e.g. successive rejection, still require large numbers of comparisons (and $\binom{p}{q}$) comparisons if q, the size of the target subset is pre-determined). The approach here is analogous to the use of L1 norms in regression, fitting a subset of items closely at the expense of downplaying the lack of fit of other items (and thus being less sensitive to outliers).

  Yes, that is correct. We have added a remark to this effect in Section 3.1.

- Choosing one 'best' model from a set of models describing the same process seems an appropriate approach. If one has two or more candidate models of different processes that give similar contributions

to z, then rather than choose one model, the appropriate conclusion is that the inversion can estimate a linear combination of these models but not distinguish between them. This could be particularly relevant for the third example, where 'lumping' of poorly distinguished weak source regions might be more appropriate than selecting a subset. The example is a good illustration of the power of the method but the method may not be the best way of inverting actual OCO-2 data.

> Thank you for this comment. We agree that the third example provides a nice illustration of the msHyBR method's ability to perform model selection in a one-step approach. In Section 4.3, we emphasize that this example highlights the applicability of our approach to very large datasets, although other approaches or additional analysis may be needed for actual data.

**Line by line**

83 Suggest using q, as above, rather than s, as number of target models to avoid confusion with s.

> Great suggestion. We have changed $s$ to $q$.

various Suggest upright T to indicate matrix-vector-transpose, since the T is not a mathematical variable. There are various online discussions on best form. My own preference would be $\mathbf{X}^{\mathsf{T}}$

> Done. All transposes have been fixed.

various The importance of this will depend on the fonts used for final production, but in the discussion preprint, the use of bold font to distinguish vectors works poorly for the Greek beta and particularly poorly for gamma. While the intended usage can be deduced from the context, it makes the article a little bit harded to read, since beta does not always appear as a vector (see eqn 7) and both forms are subscripted, once by component index within the vector, and otherwise with subscript to the vector indicating the iteration number.

> We are using the Copernicus Publications Manuscript Preparation Template for LaTeX Submissions, so it is likely that the fonts will remain. To help with clarity, we changed all $\boldsymbol{\gamma}$ to $\mathbf{g}$. We also noted that $\gamma$ was being used to refer to the noise level.
>
> Regarding $\boldsymbol{\beta}$, we decided to keep this notation to match the model selection literature. However, we clarify that $\beta_j$ corresponds to the components of the vector $\boldsymbol{\beta}$ on pages 7 and 10.

259 Strictly, $t_{k+1,k}$ and $m_{k+1,k}$ seem not to be defined at this point. Needs forward reference to Algorithm 1.

> We have clarified that these are normalization scalars and include a forward reference to the Algorithm. We also clarify that these are the elements of the upper Hessenberg matrix $\mathbf{M}$ and upper triangular matrix $\mathbf{T}$ respectively.

**Terms where a few words of description might help**

- Krylov subspace methods - Methods for working with subspaces of large problems

- Laplace distribution - Also known as double exponential.

In Section 3.2 (the first instance of the word Krylov except for the abstract), we added the statement, "Solving such optimization problems can be challenging, and Krylov subspace methods, which are iterative approaches based on Krylov subspace projections, are ideal for working in subspaces for large-scale problems."

In Section 3.1, we clarified that the Laplace distribution is also known as the double exponential.

**Notation**

- $k$ Iteration count in inversion algorithm. Dimension of the Krylov subspace at that step?? m Number of observations (dimensionality of z).

- $n$ Number of quantities to be estimated (dimensionality of s).

- $p$ Number of candidate predictors (models).

- $z$ Vector of observations that are to be fitted by the inversion.

- $s$ Vector of quantities to be estimated by the inversion.

- $\zeta$ 'Random' component of s, assumed to be distributed as zero mean multivariate Gaussian described by Q

- $\beta$ Vector of contributions from each of the candidate models, where model selection corresponds to taking zero as the estimate of particular components. The selection is 'controlled' by the regularisation parameter $\alpha$.

- $\epsilon$ Error component of z. Assumed to be zero mean multivariate Gaussian described by R.

- $R$ Covariance matrix for $\epsilon$, the error component of observations, z . H Mapping from s to observations z.

- $\lambda$ Regularisation parameter, estimated as part of the invesion, applied as scale factor of Q .

- $\alpha$ Regularisation parameter, estimated in the inversion, applied Y as defining the scale of the Laplace distribution of $\beta$.

- $\gamma$ Transformation such that minimisation over s, $\beta$ becomes minimisation over $\gamma$, $\beta$, avoiding the need to invert Q.

- $\mathbf{u}_k, \mathbf{v}_k$ Additions to the solution subspace after step k.

- $X$ Mapping from candidate models, $\beta$, to $\mathbf{s}$.

- $j$ Component index within a vector.

Thank you for this suggestion and for your list of terms. We agree that such a table can be very useful and may help with any confusion. Thus, in the revision we have included a list in Appendix A that builds on your above list. Thanks!

---

## Author Comment (AC2)

**Geoscientific Model Development**
**Reply to Anonymous Referee 2**

"A Joint Reconstruction and Model Selection Approach for Large Scale Inverse Modeling"

by Malena Sabate Landman, Julianne Chung, Jiahua Jiang, Arvind K. Saibaba, and Scot M. Miller

September 25, 2024

> We are grateful to the referee for their careful reading and thoughts on our manuscript. Below, we repeat the remarks and interleave our responses. All modifications in the revised manuscript are highlighted in **blue** and all references to equations, pages, lines, and citations correspond to the revised manuscript.

In this manuscript, the authors propose a modelling and algorithmic approach for performing model selection in large-scale inverse problems. The context is that a set of predictor variables (say, meteorological variables) are used in a linear model to predict an unknown quantity s (say, surface fluxes). The unknown quantity s is related to the observations through a forward model. Interest is on both estimating s and on identifying a small subset of the predictor variables that are most informative of s. To achieve this, the authors propose a sparsity-promoting prior on the coefficients of the predictors, and develop a hybrid iterative projection method based on flexible Krylov subspace methods for efficient optimization.

The paper is well written and the novelty and significance of the work is well justified. The paper is well organized and the figures are clear and informative. The authors perform extensive simulation studies to validate their method. My comments are mostly minor and relate to the (lack of) discussion of prior work in the statistical literature on model selection, as well as some minor points regarding the simulation studies.

My detailed comments are as follows:

1. The proposed model structure in (7) places a Laplace prior on $\beta$, the coefficients of the predictors. This is a common choice for promoting sparsity in the coefficients, which is often called the Bayesian LASSO (Park and Casella, 2008). It would be worthwhile for the authors to discuss this connection: indeed this present manuscript could be seen as the extension of the Bayesian LASSO into the inversion context. Park and Casella (2008) also address the problem of performing Markov chain Monte Carlo for this model so it may also be worth mentioning in the Conclusion where uncertainty quantification is discussed.

> We have clarified these connections and included the above reference in Section 3.1 and the conclusions.

2. Relatedly, there is literature discussing the implications of the choice of sparsity promoting prior. For example, Carvalho et al. (2010), in proposing an alternative sparsity promoting prior, discuss the limitations of the Laplace prior and other choices, and Piironen and Vehtari (2017) discuss ways to tune sparsity promoting priors. While an extensive discussion of these choices is beyond the scope of this paper, a brief mention of how these modelling choices can impact the results would be useful.

> Thanks for these references. We have included these references in a discussion in Section 3.1.

3. I was a bit confused by the one-dimension simulation example in Section 4.1. Here are my questions, which I suggest the authors clarify in the manuscript:

> Thank you very much for pointing this out. There was indeed an error on the plots, which we understand made it confusing. This has been corrected now without affecting the message of the example, which we hope is clear now.

   (a) In Figures 2 and 3, what is the x-axis showing?

   > For the 1D image deblurring example, the true signal and the observed blurred signal are defined on an equi-distant grid on $[0, 1]$ containing 100 points. A quadrature method is used to discretize the continuous linear inverse problem,
   >
   > $$\int_0^1 h(s-t)f(t)dt = g(s), \quad 0 \le s \le 1.$$
   >
   > Thus, the x-axis denotes the domain where the signal and measurements are defined.
   >
   > We have now clarified this in the caption of Figure 2.

   (b) In Figure 2(b), can the authors show $s$ and $X\beta$ as well as the blurred observation?

   > We have now added a subfigure, Figure 2 (c), showing the true solution as well as its components.

   (c) In Figure 3(a), I was surprised to see that all methods noticeably overestimate the true function, particularly for the x-axis values between 0 and 0.5. Shouldn't the stochastic component, $\zeta$, correct this? How does this error arise?

   > Thank you very much for this comment. This was indeed an error on the plot, where we had plotted the mean rather than the full true solution. We have now updated all figures affected (Figure 3 and Figure 5).

   (d) What exactly is the "reconstruction error norm" and the "relative reconstruction error norm" shown in Figure 5 and mentioned in the text (also in the later simulation studies)?

   > In Section 4.1.1, we have defined the relative reconstruction error norm and fixed all instances to be consistent.

4. In discussing the partial F-test in Section 2, it may be worth mentioning that the "smaller" model must nest within the "larger" model, a notable limitation of the F-test compared to competing methods.

> We have edited the corresponding paragraph of Sect. 2 to highlight this point.

5. Minor points:

   (a) Lines 419–420: the references Miller et al and Liu et al would be better parenthesised.

   > Done.

(b) Lines 485–486: I'm surprised that the elements of Q have units (ppm)2—isn't this quantity a flux?

Thanks for catching this. It has been fixed.

**References**

- Carvalho, C. M., Polson, N. G., and Scott, J. G. (2010). The horseshoe estimator for sparse signals. Biometrika, 97(2):465–480.

- Park, T. and Casella, G. (2008). The Bayesian Lasso. Journal of the American Statistical Association, 103(482):681–686.

- Piironen, J. and Vehtari, A. (2017). Sparsity information and regularization in the horseshoe and other shrinkage priors. Electronic Journal of Statistics, 11(2):5018–5051.

---

## Author Comment (AC3)

**Geoscientific Model Development**
**Reply to topic editor**

"A Joint Reconstruction and Model Selection Approach for Large Scale Inverse Modeling"
by Malena Sabate Landman, Julianne Chung, Jiahua Jiang, Arvind K. Saibaba, and Scot M. Miller

September 25, 2024

Dear authors,
Thank you for submitting your manuscript to GMD. While we are currently looking for reviewers, there is one additional point I partly overlooked in the initial assessment process, namely, the fact that it would be valuable to have the software name and version you used to implement your novel algorithm to be more explicitly stated in the introduction, as well as in the title (as per GMD guidelines).

Thanks so much for your message and for your detailed work in handling our submission.
As requested, we have added the software and algorithm name to the title of the paper and included a statement in the in introduction of the paper.

Please address this point in a later stage, when working on addressing referees' comments.
Thank you.
Best regards, Ludovic Räss